# QuSpin: a Python package for dynamics and exact diagonalisation of quantum many body systems Part II: bosons, fermions and higher spins

**Phillip Weinberg[1⋆] and Marin Bukov[2]**

**1** Department of Physics, Boston University,
590 Commonwealth Ave., Boston, MA 02215, USA
**2** Department of Physics, University of California Berkeley, CA 94720, USA

⋆ weinbe58@bu.edu

## Abstract

**We present a major update to QuSpin, *SciPostPhys.2.1.003* – an open-source Python package for exact diagonalization and quantum dynamics of arbitrary boson, fermion and spin many-body systems, supporting the use of various (user-defined) symmetries in one and higher dimension and (imaginary) time evolution following a user-specified driving protocol. We explain how to use the new features of QuSpin using seven detailed examples of various complexity: (i) the transverse-field Ising chain and the Jordan-Wigner transformation, (ii) free particle systems: the Su-Schrieffer-Heeger (SSH) model, (iii) the many-body localized 1D Fermi-Hubbard model, (iv) the Bose-Hubbard model in a ladder geometry, (v) nonlinear (imaginary) time evolution and the Gross-Pitaevskii equation on a 1D lattice, (vi) integrability breaking and thermalizing dynamics in the translationally-invariant 2D transverse-field Ising model, and (vii) out-of-equilibrium Bose-Fermi mixtures. This easily accessible and user-friendly package can serve various purposes, including educational and cutting-edge experimental and theoretical research. The complete package documentation is available under http://weinbe58.github.io/QuSpin/index.html.**

## 1 What Problems can I Study with QuSpin?

Understanding the physics of many-body quantum condensed matter systems often involves a great deal of numerical simulations, be it to gain intuition about the complicated problem of interest, or because they do not admit an analytical solution which can be expressed in a closed form. This motivates the development of open-source packages [3–15], the purpose of which is to facilitate the study of condensed matter systems, without the need to implement the inner workings of complicated numerical methods which required years to understand and fully develop. Here, we report on a major upgrade to QuSpin [16] – a Python library for exact diagonalisation (ED) and simulation of the dynamics of arbitrary quantum many-body systems.

Although ED methods are vastly outperformed by more sophisticated numerical techniques in the study of equilibrium problems, such as quantum Monte Carlo [17–19], matrix product states based density matrix renormalisation group [20–22], and dynamical mean-field theory [23–25], as of present date ED remains essential for certain dynamical non-equilibrium problems. The reason for this often times relies on the fact that the underlying physics of these problems cannot be explained without taking into consideration the contribution from high-energy states excited during the nonequilibrium process. Some prominent examples of such problems include the study of the many-body localisation (MBL) transition [26–30], the Eigenstate Thermalisation hypothesis [31], ergodicity breaking, thermalization and scrambling [32–34], quantum quench dynamics [35], periodically-driven systems [36–44], non-demolition measurements in many-body systems [45], long-range quantum coherence [46], dynamics-induced instabilities [47–54], adiabatic and counter-diabatic state preparation [55–59], dynamical [60, 61] and topological [62] phase transitions applications of Machine Learning to (non-equilibrium) physics [51, 63–68], optimal control [69–71], to benchmark results in sophisticated algorithms, such as quantum annealing [72] and Quantum Monte Carlo [73], and many more. Besides dynamical studies, ED methods are currently heavily used to compute the full (or low-energy) spectrum of frustrated Hamiltonians, where they compete with a high-precision tensor network approach. Last but not least, virtually all new numerical techniques are frequently benchmarked against ED.

It is, thus, arguably useful to have a toolbox available at one's disposal which allows one to quickly simulate and study these and related nonequilibrium problems. As such, QuSpin

offers easy access to performing numerical simulations, which can facilitate the development and inspiration of new ideas and the discovery of novel phenomena, eliminating the cost of spending time to develop a reliable code. Besides theorists, the new version of QuSpin will hopefully even prove valuable to experimentalists working on problems containing dynamical setups, as it can help students and researchers focus on perfecting the experiment, rather than worry about writing the supporting simulation code. Last but not least, with the computational processing power growing higher than ever before, the role played by simulations for theoretical research becomes increasingly more important too. It can, therefore, be expected that in the near future quantum simulations become an integral part of the standard physics university curriculum, and having easily accessible toolboxes, such as QuSpin, is one of the requisites for this anticipated change.

# 2 How do I Use the New Features of QuSpin?

New in QuSpin 3.0, we have added the following features and toolboxes:

(i) support for fermion, boson and higher-spin Hamiltonians with `basis.ent_entropy` and `basis.partial_trace` routine to calculate entanglement entropy, reduced density matrices, and entanglement spectrum.

(ii) `general` basis constructor classes for user-defined symmetries which allow the implementation of higher-dimensional lattice structures.

(iii) `block_ops` class and `block_diag_hamiltonian` function to automatically handle splitting the evolution over various symmetry sectors of the Hilbert space.

(iv) user-customisable `evolve` routine to handle user-specified linear and non-linear equations of motion.

(v) `quantum_operator` class to define parameter-dependent Hamiltonians.

(vi) `quantum_LinearOperator` class which applies the operator "on the fly" which saves vast amounts of memory at the cost of computational time.

Before we carry on, we refer the interested reader to Examples 0-3 from the original QuSpin paper [16]. The examples below focus predominantly on the newly introduced features, and are thus to be considered complementary. We emphasize that, since they serve the purpose of explaining how to use QuSpin, for the sake of brevity we shall not discuss the interesting physics related to the interpretation of the results.

*Installing QuSpin is quick and efficient; just follow the steps outlined in App. A.*

## 2.1 The Spectrum of the Transverse Field Ising Model and the Jordan-Wigner Transformation

This example shows how to

- construct fermionic hopping, $p$-wave pairing and on-site potential terms, and spin$-1/2$ interactions and transverse fields,
- implement periodic and anti-periodic boundary conditions,
- use particle conservation modulo 2, and spin inversion,

- handle the default built-in particle conservation and symmetry checks,
- obtain the spectrum of a QuSpin Hamiltonian.

*Physics Setup*—The transverse field Ising (TFI) chain is paradigmatic in our understanding of quantum phase transitions, since it represents an exactly solvable model [74]. The Hamiltonian is given by

$$H=\sum_{j=0}^{L-1}-J\sigma^z_{j+1}\sigma^z_j-h\sigma^x_j, \tag{1}$$

where the nearest-neighbour (nn) spin interaction is $J$, $h$ denotes the transverse field, and $\sigma^\alpha_j$ are the Pauli spin-1/2 matrices. We use periodic boundary conditions and label the $L$ lattice sites $0,\dots,L-1$ to conform with Python's convention. This model has gapped, fermionic elementary excitations, and exhibits a phase transition from an antiferromagnet to a paramagnet at $(h/J)_c=1$. The Hamiltonian possesses the following symmetries: parity (reflection w.r.t. the centre of the chain), spin inversion, and (many-body) momentum conservation.

In one dimension, the TFI Hamiltonian can be mapped to spinless $p$-wave superconducting fermions via the Jordan-Wigner (JW) transformation [74–76]:

$$c_i=\frac{\sigma^x_i-i\sigma^y_i}{2}\prod_{j<i}\sigma^z_j,\qquad c^\dagger_i=\frac{\sigma^x_i+i\sigma^y_i}{2}\prod_{j<i}\sigma^z_j, \tag{2}$$

where the fermionic operators satisfy $\{c_i,c^\dagger_j\}=\delta_{ij}$. The Hamiltonian is readily shown to take the form

$$H=\sum_{j=0}^{L-1}J\left(-c^\dagger_jc_{j+1}+c_jc^\dagger_{j+1}\right)+J\left(-c^\dagger_jc^\dagger_{j+1}+c_jc_{j+1}\right)+2h\left(n_j-\frac{1}{2}\right). \tag{3}$$

In the fermionic representation, the spin $zz$-interaction maps to nn hopping and a $p$-wave pairing term with coupling constant $J$, while the transverse field translates to an on-site potential shift of magnitude $h$. In view of the implementation of the model using QuSpin, we have ordered the terms such that the site index is growing to the right, which comes at the cost of a few negative signs due to the fermion statistics. We emphasize that this ordering is not required by QuSpin, but it is merely our choice to use it here for the sake of consistency. The fermion Hamiltonian posses the symmetries: particle conservation modulo 2, parity and (many-body) "momentum" conservation.

Here, we are interested in studying the spectrum of the TFI model in both the spin and fermion representations. However, if one naïvely carries out the JW transformation, and computes the spectra of Eqs. (1) and (3), one might be surprised that they do not match exactly. The reason lies in the form of the boundary condition required to make the JW mapping exact – a subtle issue often left aside in favour of discussing the interesting physics of the TFI model itself.

We recall that the starting point is the periodic boundary condition imposed on the spin Hamiltonian in Eq. (1). Due to the symmetries of the spin Hamiltonian (1), we can define the JW transformation on every symmetry sector separately. To make the JW mapping exact, we supplement Eq. (2) with the following boundary conditions: (i) the negative spin-inversion symmetry sector maps to the fermion Hamiltonian (3) with *periodic* boundary conditions (PBC) and *odd* total number of fermions; (ii) the positive spin-inversion symmetry sector maps to the fermion Hamiltonian (3) with *anti-periodic* boundary conditions (APBC) and *even* total number of fermions. Anti-periodic boundary conditions differ from PBC by a negative sign attached to all coupling constants that cross a single, fixed lattice bond (the bond itself is arbitrary as

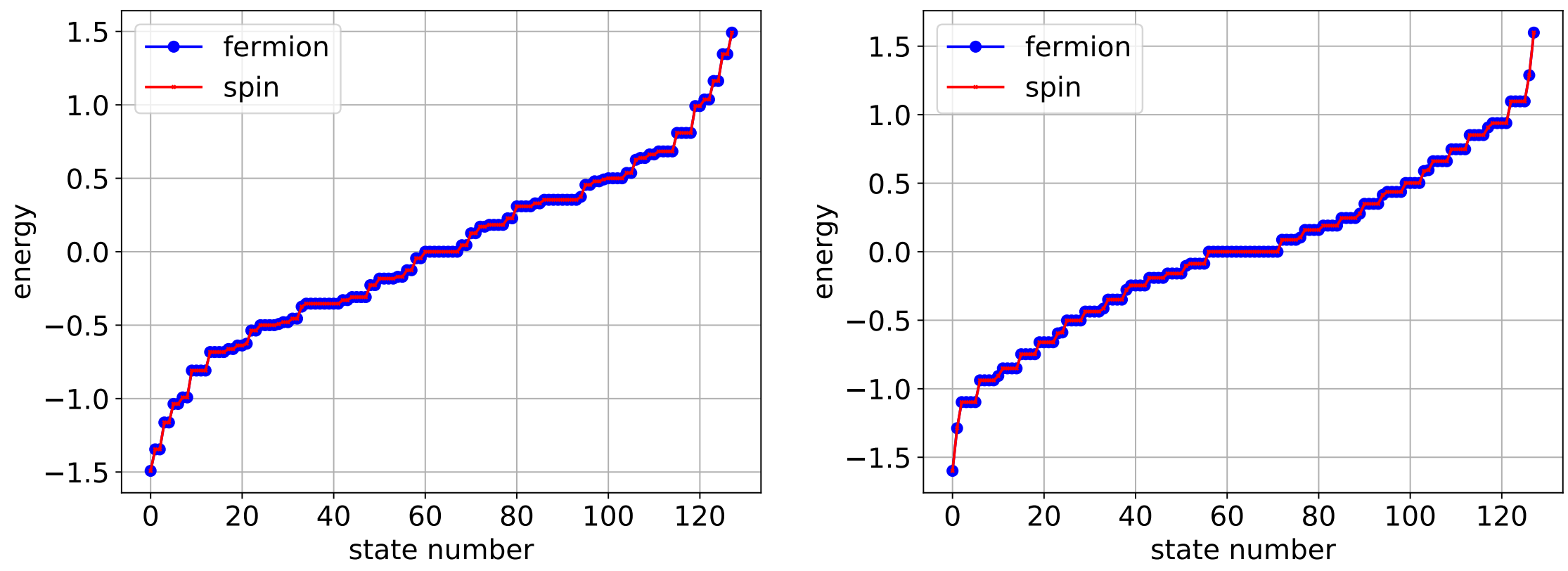

(a) negative spin inversion/PBC sector

(b) positive spin inversion/APBC sector

Figure 1: Comparison of the spectra [in units of $J$] of the spin (1) and fermion (3) representation of the transverse field Ising Hamiltonian. The degeneracy in the spectrum is due to the remaining reflection and translation symmetries which could easily be taken into account (see text). The parameters are $J = 1.0$, $h = \sqrt{2}$, and $L = 8$.

all bonds are equal for PBC). APBC and PBC are special cases of the more general twisted boundary conditions where, instead of a negative sign, one attaches an arbitrary phase factor.

In the following, we show how to compute the spectra of the Hamiltonians in Eqs. (1) and (3) with the correct boundary conditions using QuSpin. Figure 1 shows that they match exactly in both the PBC and APBC cases discussed above, as predicted by theory.

*Code Analysis*—We begin by loading the QuSpin operator and basis constructors, as well as some standard Python libraries.

```
from quspin.operators import hamiltonian # Hamiltonians and operators
from quspin.basis import spin_basis_1d, spinless_fermion_basis_1d # Hilbert space
    spin basis
import numpy as np # generic math functions
import matplotlib.pyplot as plt # plotting library
```

First, we define the models parameters.

```
##### define model parameters #####
L=8 # system size
J=1.0 # spin zz interaction
h=np.sqrt(2) # z magnetic field strength
```

We have to consider two cases when computing the spectrum, as discussed in the theory section above. In one case, the fermionic system has PBC and the particle number sector is odd, while the spins are constrained to the negative spin inversion symmetry sector, while in the second – the fermion model has APBC with even particle number sector, and the spin model is considered in the positive spin inversion sector. To this end, we introduce the variables `zblock` $\in \{\pm 1\}$ and PBC $\in \{\pm 1\}$, where PBC $= -1$ denotes APBC. Note that the only meaningful combinations are (`zblock`, PBC) $= (-1, 1), (1, -1)$, which we loop over:

```
# loop over spin inversion symmetry block variable and boundary conditions
for zblock,PBC in zip([-1,1],[1,-1]):
```

Within this loop, the code is divided in two independent parts: first, we compute the spectrum of the TFI system, and then – that of the equivalent fermionic model. Let us discuss the spins.

In QuSpin, operators are stored as sparse lists. These lists contain two parts: (i) the lattice sites on which the operator acts together with the coupling strength, which we call a

*site-coupling* list, and (ii) the types of the operators involved, i.e. the *operator-string*. For example, the operator $\mathcal{O} = g\sum_{j=0}^{L-1}\sigma_j^\mu$ can be uniquely represented by the site-coupling list `[[g,0],[g,1],...,[g,L-1]]`, and the information that it is the Pauli matrix $\mu$. The components lists are nothing but the pairs of the field strength and the site index `[g,j]`. It is straightforward to generalise this to non-uniform fields `g` $\mapsto$ `g[j]`. Similarly, any two-body operator $\mathcal{O} = J_{zz}\sum_{j=0}^{L-1}\sigma_j^\mu\sigma_{j+1}^\nu$ can be fully represented by the two sites it acts on, and its coupling strength: `[J,j,j+1]`. We then stack up these elementary lists together into the full site-coupling list: `[[J,0,1],[J,1,2],...,[J,L-2,L-1],[J,L-1,0]]`.

```
    ##### define spin model
    # site-coupling lists (PBC for both spin inversion sectors)
    h_field=[[-h,i] for i in range(L)]
    J_zz=[[-J,i,(i+1)%L] for i in range(L)] # PBC
```

Notice the way we defined the periodic boundary condition for the spin-spin interaction using the modulo operator `%`, which effectively puts a coupling between sites $L-1$ and 0. We mention in passing that the above procedure generalises so one can define any multi-body local or nonlocal operator using QuSpin.

In order to specify the types of the on-site single-particle operators, we use operator strings. For instance, the transverse field operator $\mathcal{O} = g\sum_{j=0}^{L-1}\sigma_j^x$ becomes `['x',h_field]`, while the two-body $zz$-interaction is `['zz',J_zz]`. It is important to notice that the order of the letters in the operator string corresponds to the order the operators are listed in the site-coupling lists. Putting everything into one final list yields the `static` list for the spin model:

```
    # define spin static and dynamic lists
    static_spin =[["zz",J_zz],["x",h_field]] # static part of H
```

In QuSpin, the user can define both static and dynamic operators. Since this example does not contain any time-dependent terms, we postpone the explanation of how to use dynamic lists to Sec. 2.5, and use an empty list instead.

```
    dynamic_spin=[] # time-dependent part of H
```

The last step before we can construct the Hamiltonian is to build the basis for it. This is done using the `basis` constructors. For spin systems, we use `spin_basis_1d` which supports the operator strings `'z','+','-','I'`, and for spin-1/2 additionally `'x','y'`. The first and required argument is the number of sites $L$. Optional arguments are used to parse symmetry sectors. For instance, if we want to construct an operator in the spin-inversion block with quantum number $+1$, we can conveniently do this using the flag `zblock=1`.

```
    # construct spin basis in pos/neg spin inversion sector depending on APBC/PBC
    basis_spin = spin_basis_1d(L=L,zblock=zblock)
```

Having specified the static and dynamic lists, as well as the basis, building up the Hamiltonian is a one-liner, using the `hamiltonian` constructor. The required arguments in order of appearance are the `static` and `dynamic` lists, respectively. Optional arguments include the `basis`, and the precision or data type `dtype`. If no basis is passed, the constructor uses `spin_basis_1d` by default. The default data type is `np.complex128`.

```
    # build spin Hamiltonians
    H_spin=hamiltonian(static_spin,dynamic_spin,basis=basis_spin,dtype=np.float64
    )
```

The Hamiltonian is stored as a Scipy sparse matrix for efficiency. It can be converted to a dense array for a more convenient inspection using the attribute `H.toarray()`. To calculate its spectrum, we use the attribute `H.eigenvalsh()`, which returns all eigenvalues. Other attributes for diagonalisation were discussed here, c.f. Ref. [16].

```
    # calculate spin energy levels
    E_spin=H_spin.eigvalsh()
```

Let us now move to the second part of the loop which defines the fermionic $p$-wave superconductor. We start by defining the site-coupling list for the local potential

```
    ##### define fermion model
    # define site-coupling lists for external field
    h_pot=[[2.0*h,i] for i in range(L)]
```

Let us focus on the case of periodic boundary conditions PBC=1 first.

```
    if PBC==1: # periodic BC: odd particle number subspace only
```

In the fermion model, we have two types of two-body terms: hopping terms $c_i^\dagger c_{i+1} - c_i c_{i+1}^\dagger$, and pairing terms $c_i^\dagger c_{i+1}^\dagger - c_i c_{i+1}$. While QuSpin allows any ordering of the operators in the static and dynamic lists, for the sake of consistency we set a convention: the site indices grow to the right. To take into account the opposite signs resulting from the fermion statistics, we have to code the site-coupling lists for all four terms separately. These two-body terms are analogous to the spin-spin interaction above:

```
        # define site-coupling lists (including boudary couplings)
        J_pm=[[-J,i,(i+1)%L] for i in range(L)] # PBC
        J_mp=[[+J,i,(i+1)%L] for i in range(L)] # PBC
        J_pp=[[-J,i,(i+1)%L] for i in range(L)] # PBC
        J_mm=[[+J,i,(i+1)%L] for i in range(L)] # PBC
```

To construct a fermionic operator, we make use of the basis constructor `spinless_fermion_basis_1d`. Once again, we pass the number of sites `L`. As we explained in the analysis above, we need to consider all odd particle number sectors in the case of PBC. This is done by specifying the particle number sector `Nf`.

```
        # construct fermion basis in the odd particle number subsector
        basis_fermion = spinless_fermion_basis_1d(L=L,Nf=range(1,L+1,2))
```

In the case of APBC, we first construct all two-body site-coupling lists as if the boundaries were open, and supplement the APBC links with negative coupling strength in the end:

```
    elif PBC==-1: # anti-periodic BC: even particle number subspace only
        # define bulk site coupling lists
        J_pm=[[-J,i,i+1] for i in range(L-1)]
        J_mp=[[+J,i,i+1] for i in range(L-1)]
        J_pp=[[-J,i,i+1] for i in range(L-1)]
        J_mm=[[+J,i,i+1] for i in range(L-1)]
        # add boundary coupling between sites (L-1,0)
        J_pm.append([+J,L-1,0]) # APBC
        J_mp.append([-J,L-1,0]) # APBC
        J_pp.append([+J,L-1,0]) # APBC
        J_mm.append([-J,L-1,0]) # APBC
```

The construction of the basis is the same, except that this time we need all even particle number sectors:

```
        # construct fermion basis in the even particle number subsector
        basis_fermion = spinless_fermion_basis_1d(L=L,Nf=range(0,L+1,2))
```

As before, we need to specify the type of operators that go in the fermion Hamiltonian using operator string lists. The `spinless_fermion_basis_1d` class accepts the following strings `'+','-','n','I'`, and additionally the particle-hole symmetrised density operator `'z'` $= n - 1/2$. The static and dynamic lists read as

```
    # define fermionic static and dynamic lists
    static_fermion =[["+-",J_pm],["-+",J_mp],["++",J_pp],["--",J_mm],['z',h_pot]]
    dynamic_fermion=[]
```

Constructing and diagonalising the fermion Hamiltonian is the same as for the spin-1/2 system. Note that one can disable the automatic built-in checks for particle conservation `check_pcon=False` and all other symmetries `check_symm=False` if one wishes to suppress the checks.

```
    # build fermionic Hamiltonian
    H_fermion=hamiltonian(static_fermion,dynamic_fermion,basis=basis_fermion,
                          dtype=np.float64,check_pcon=False,check_symm=False)
    # calculate fermionic energy levels
    E_fermion=H_fermion.eigvalsh()
```

The complete code including the lines that produce Fig. 1 is available under:

http://weinbe58.github.io/QuSpin/examples/example4.html

## 2.2 Free Particle Systems: the Fermionic Su-Schrieffer-Heeger (SSH) Chain

This example shows how to

- construct free-particle Hamiltonians in real space,
- implement translation invariance with a two-site unit cell and construct the single-particle momentum-space block-diagonal Hamiltonian using the `block_diag_hamiltonian` tool,
- compute non-equal time correlation functions,
- time-evolve multiple quantum states simultaneously.

*Physics Setup*—The Su-Schrieffer-Heeger (SSH) model of a free-particle on a dimerised chain was first introduced in the seventies to study polyacetylene [77–79]. Today, this model is paradigmatically used in one spatial dimension to introduce the concepts of Berry phase, topology, edge states, etc [80]. The Hamiltonian is given by

$$H = \sum_{j=0}^{L-1} -(J + (-1)^j \delta J)\left(c_j c_{j+1}^\dagger - c_j^\dagger c_{j+1}\right) + \Delta(-1)^j n_j, \tag{4}$$

where $\{c_i, c_j^\dagger\} = \delta_{ij}$ obey fermionic commutation relations. The uniform part of the hopping matrix element is $J$, the bond dimerisation is defined by $\delta J$, and $\Delta$ is the staggered potential. We work with periodic boundary conditions.

Below, we show how one can use QuSpin to study the physics of free fermions in the SSH chain. One way of doing this would be to work in the many-body (Fock space) basis, see Sec. 2.3. However, whenever the particles are non-interacting, the exponential scaling of the Hilbert space dimension with the number of lattice sites imposes an artificial limitation on the system sizes one can do. Luckily, with no interactions present, the many-body wave functions factorise in a product of single-particle states. Hence, it is possible to study the behaviour of many free bosons and fermions by simulating the physics of a single particle, and populating the states according to bosonic or fermionic statistics, respectively.

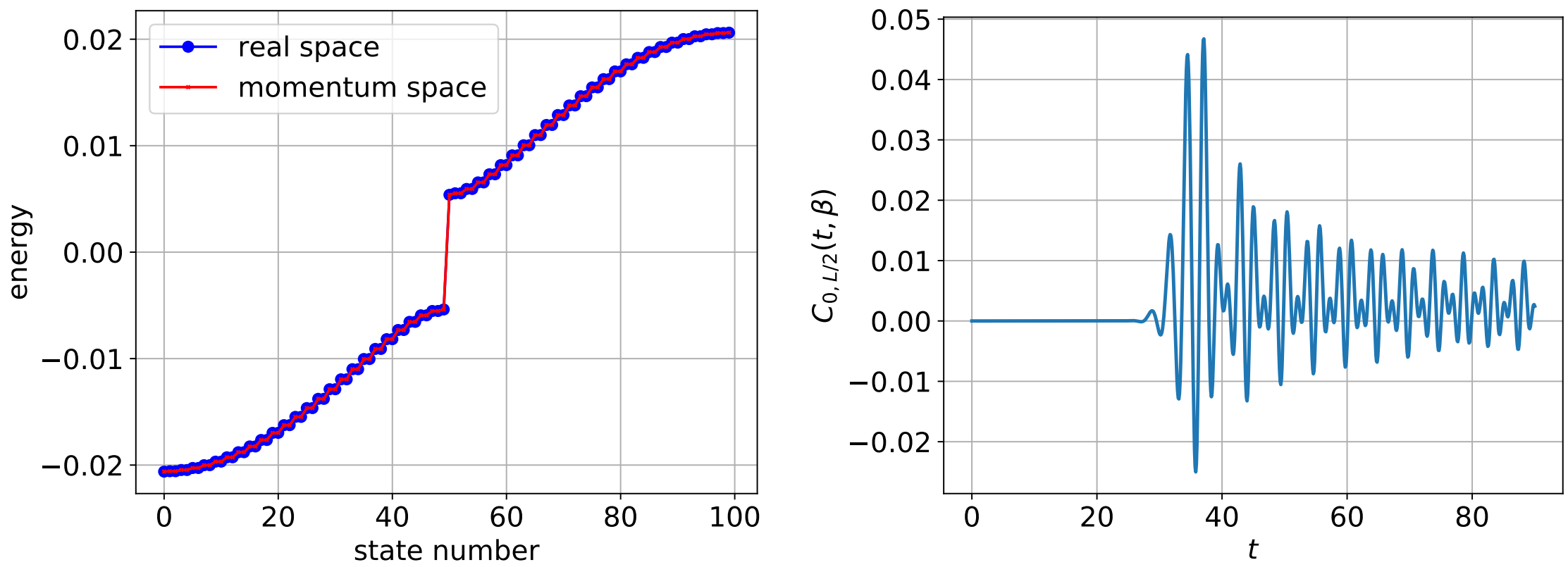

Figure 2: (a) the spectrum [in units of $J$] of the SSH Hamiltonian in real and momentum space. (b) The non-equal time correlator $C_{i=0,j=L/2}(t)$, cf. Eq. (6) as a function of time. The parameters are $\delta J/J = 0.1$, $\Delta/J = 0.5$, $J\beta = 100.0$ and $L = 100$.

Making use of translation invariance, a straightforward Fourier transformation to momentum space, $a_k = \sqrt{2/L}\sum_{j:\text{even}}^{L-1} \mathrm{e}^{-ikj} c_j$ and $b_k = \sqrt{2/L}\sum_{j:\text{odd}}^{L-1} \mathrm{e}^{-ikj} c_j$, casts the SSH Hamiltonian in the following form

$$H = \sum_{k\in\mathrm{BZ}'} (a_k^\dagger, b_k^\dagger)\begin{pmatrix} \Delta & -(J+\delta J)\mathrm{e}^{-ik} - (J-\delta J)\mathrm{e}^{+ik} \\ -(J+\delta J)\mathrm{e}^{+ik} - (J-\delta J)\mathrm{e}^{-ik} & -\Delta \end{pmatrix}\begin{pmatrix} a_k \\ b_k \end{pmatrix}, \tag{5}$$

where the reduced Brillouin zone is defined as $\mathrm{BZ}' = [-\pi/2, \pi/2)$. We thus see that the Hamiltonian reduces further to a set of independent $2\times 2$ matrices. The spectrum of the SSH model is gapped and, thus, has two bands, see Fig. 2a.

Since we are dealing with free fermions, the ground state is the Fermi sea, $|\mathrm{FS}\rangle$, defined by filling up the lowest band completely. We are interested in measuring the real-space non-equal time density autocorrelation function

$$C_{ij}(t) = \langle \mathrm{FS}|n_i(t)n_j(0)|\mathrm{FS}\rangle = \langle \mathrm{FS}(t)|n_i(0)\underbrace{U(t,0)n_j(0)|\mathrm{FS}\rangle}_{|\mathrm{nFS}(t)\rangle}. \tag{6}$$

To evaluate the correlator numerically, we shall use the right-hand side of this equation.

As we are studying free particles, it is enough to work with the single-particle states. For instance, the Fermi sea can be obtained as $|\mathrm{FS}\rangle = \prod_{k\le k_F} c_k^\dagger|0\rangle$. Denoting the on-site density operator by $n_i$, one can cast the correlator in momentum space in the following form:

$$C_{ij}(t) = \sum_{k\le k_F} \langle k|n_i(t)\hat{n}_j(0)|k\rangle. \tag{7}$$

If we want to consider finite-temperature $\beta^{-1}$, the above formula generalises to

$$C_{ij}(t,\beta) = \sum_{k\in\mathrm{BZ}'} n_{\mathrm{FD}}(k,\beta)\langle k|n_i(t)n_j(0)|k\rangle, \tag{8}$$

where $n_{\mathrm{FD}}(k,\beta) = 1/(\exp(\beta E_k)+1)$ is the Fermi-Dirac distribution at temperature $\beta^{-1}$, with $E_k$ the SSH dispersion. Figure 2b) shows the time evolution of $C_{ij}(t,\beta)$ for two sites, separated by the maximal distance on the ring: $L/2$.

*Code Analysis.*—Let us explain how one can do all this quickly and efficiently using QuSpin. As always, we start by loading the required packages and libraries.

```
from quspin.operators import hamiltonian,exp_op # Hamiltonians and operators
from quspin.basis import spinless_fermion_basis_1d # Hilbert space fermion basis
from quspin.tools.block_tools import block_diag_hamiltonian # block diagonalisation
import numpy as np # generic math functions
import matplotlib.pyplot as plt # plotting library
try: # import python 3 zip function in python 2 and pass if already using python 3
    import itertools.izip as zip
except ImportError:
    pass
```

After that, we define the model parameters

```
##### define model parameters #####
L=100 # system size
J=1.0 # uniform hopping
deltaJ=0.1 # bond dimerisation
Delta=0.5 # staggered potential
beta=100.0 # inverse temperature for Fermi-Dirac distribution
```

In the following, we construct the fermionic SSH Hamiltonian first in real space. We then show how one can also construct it in momentum space where, provided we use periodic boundary conditions, it appears bock-diagonal. Let us define the fermionic site-coupling lists. Once again, we emphasise that fermion systems require special care in defining the hopping terms: Eq. (4) is conveniently cast in the form where all site indices on the operators grow to the right, and all signs due to the fermion statistics are explicitly spelt out.

```
##### construct single-particle Hamiltonian #####
# define site-coupling lists
hop_pm=[[-J-deltaJ*(-1)**i,i,(i+1)%L] for i in range(L)] # PBC
hop_mp=[[+J+deltaJ*(-1)**i,i,(i+1)%L] for i in range(L)] # PBC
stagg_pot=[[Delta*(-1)**i,i] for i in range(L)]
```

To define the `static` list we assign the corresponding operator strings to he site-coupling lists. Since our problem does not feature any explicit time dependence, we leave the `dynamic` list empty.

```
# define static and dynamic lists
static=[["+-",hop_pm],["-+",hop_mp],['n',stagg_pot]]
dynamic=[]
```

Setting up the fermion basis with the help of the constructor `spinless_fermion_basis_1d` proceeds as smoothly as in Sec. 2.1. Notice a cheap trick: by specifying a total of `Nf=1` fermion in the lattice, QuSpin actually allows to define single-particle models, as a special case of the more general many-body Hamiltonians. Compared to many body models, however, due to the exponentially reduced Hilbert space size, this allows us to scale up the system size `L`.

```
# define basis
basis=spinless_fermion_basis_1d(L,Nf=1)
```

We then build the real-valued SSH Hamiltonian in real space by passing the static and dynamic lists, as well as the basis and the data type. After that, we and diagonalise it, storing all eigenenergies and eigenstates.

```
# build real-space Hamiltonian
H=hamiltonian(static,dynamic,basis=basis,dtype=np.float64)
# diagonalise real-space Hamiltonian
E,V=H.eigh()
```

For translation invariant single-particle models, however, the user might prefer to use momentum space, where the Hamiltonian becomes block diagonal, as we showed in the theory section above. This can be achieved using QuSpin's `block_tools`. The idea behind this tool is simple: the main purpose is to create the full Hamiltonian in block-diagonal form, where the blocks correspond to pre-defined quantum numbers. In our case, we would like to use momentum or `kblock`'s. Note that the unit cell in the SSH model contains two sites, which we encode using the variable `a=2`. Thus, we can create a list of dictionaries `blocks`, each element of which defines a single symmetry block. If we combine all blocks, we exhaust the full Hilbert space. All other basis arguments, such as the system size, we store in the variable `basis_args`. We invite the interested user to check the package documentation for additional optional arguments and functionality of `block_tools`, cf. App. C. We mention in passing that this procedure is independent of the symmetry, and can be done using all symmetries supported by QuSpin, not only translation.

```
##### compute Fourier transform and momentum-space Hamiltonian #####
# define momentm blocks and basis arguments
blocks=[dict(Nf=1,kblock=i,a=2) for i in range(L//2)] # only L//2 distinct
    momenta
basis_args = (L,)
```

To create the block-diagonal Hamiltonian, we invoke the `block_diag_hamiltonian` method. It takes both required and optional arguments, and returns the transformation which block-diagonalises the Hamiltonian (in our case the Fourier transform) and the block-diagonal Hamiltonian object. Required arguments, in order of appearance, are the `blocks`, the `static` and `dynamic` lists, the `basis` constructor, `basis_args`, and the data type. Since we anticipate the matrix elements of the momenum-space Hamiltonian to contain the Fourier factors $\exp(-ik)$, we know to choose a complex data type. `block_diag_hamiltonian` also accepts some optional arguments, such as the flags for disabling the automatic built-in symmetry checks. More about this function can be found in the documentation, cf. App. C.

```
# construct block-diagonal Hamiltonian
FT,Hblock = block_diag_hamiltonian(blocks,static,dynamic,
    spinless_fermion_basis_1d,
                    basis_args,np.complex128,get_proj_kwargs=dict(pcon=True))
```

We can then use all functions and methods of the `hamiltonian` class to manipulate the block-diagonal Hamiltonian, for instance the diagonalisation routines:

```
# diagonalise momentum-space Hamiltonian
Eblock,Vblock=Hblock.eigh()
```

Last, we proceed to calculate the correlation function from Eq. (6). To this end, we shall split the correlator according to the RHS of Eq. (6). Thus, the strategy is to evolve both the Fermi sea $|\mathrm{FS}(t)\rangle$ and the auxiliary state $|\mathrm{nFS(t)}\rangle$ in time, and compute the expectation value of the time-independent operator $n_i(0)$ in between the two states as a function of time. Keep in mind that we do not need to construct the Fermi sea as a many-body state explicitly, so we rather work with single-particle states.

The first step is to collect all momentum eigenstates into the columns of the array `psi`. We then build the operators $n_{j=0}$ and $n_{j=L/2}$ in real space.

```
##### prepare the density observables and initial states #####
# grab single-particle states and treat them as initial states
psi0=Vblock
# construct operator n_1 = $n_{j=0}$
n_1_static=[['n',[[1.0,0]]]]
n_1=hamiltonian(n_1_static,[],basis=basis,dtype=np.float64,
                check_herm=False,check_pcon=False)
```

```
# construct operator n_2 = $n_{j=L/2}$
n_2_static=[['n',[[1.0,L//2]]]]
n_2=hamiltonian(n_2_static,[],basis=basis,dtype=np.float64,
                check_herm=False,check_pcon=False)
```

Next, we transform these two operators to momentum space using the method `rotate_by()`. Setting the flag `generator=False` treats the Fourier transform `FT` as a unitary transformation, rather than a generator of a unitary transformation.

```
# transform n_j operators to momentum space
n_1=n_1.rotate_by(FT,generator=False)
n_2=n_2.rotate_by(FT,generator=False)
```

Let us proceed with the time-evolution. We first define the time vector `t` and the state `n_psi0`.

```
##### evaluate nonequal time correlator <FS|n_2(t) n_1(0)|FS> #####
# define time vector
t=np.linspace(0.0,90.0,901)
# calcualte state acted on by n_1
n_psi0=n_1.dot(psi0)
```

We can perform the time evolution in two ways: (i) we calculate the time-evolution operator `U` using the `exp_op` class [1], and apply it to the momentum states `psi0` and `n_psi0`. The `exp_op` class calculates the matrix exponential $\exp(aH)$ of an operator $H$ multiplied by a complex number $a$. One can also conveniently compute a series of matrix exponentials $\exp(aHt)$ for every time $t$ by specifying the stating point `start`, endpoint `stop` and the number of elements `num` which define the time array $t$ via `t=np.linspace(start,stop,num)`. Last, by parsing the flag `iterate=True` we create a python generator – a pre-defined object evaluated only at the time it is called, i.e. not pre-computed, which can save both time and memory.

```
# construct time-evolution operator using exp_op class (sometimes faster)
U = exp_op(Hblock,a=-1j,start=t.min(),stop=t.max(),num=len(t),iterate=True)
# evolve states
psi_t=U.dot(psi0)
n_psi_t = U.dot(n_psi0)
```

Another way of doing the time evolution, (ii), is to use the `evolve()` method of the `hamiltonian` class. The idea here is that every Hamiltonian defines a generator of time translations. This method solves the Schrödinger equation using SciPy's ODE integration routines, see App. C for more details. The required arguments, in order of appearance, are the initial state, the initial time, and the time vector. The `evolve()` method also supports the option to create the output as a generator using the flag `iterate=True`. Both ways (i) and (ii) time-evolve all momentum states `psi` at once, i.e. simultaneously.

```
# alternative method for time evolution using Hamiltonian class
#psi_t=Hblock.evolve(psi0,0.0,t,iterate=True)
#n_psi_t=Hblock.evolve(n_psi0,0.0,t,iterate=True)
```

To evaluate the correlator, we first preallocate memory by defining the empty array `correlators`, which will be filled with the correlator values in every single-particle momentum mode $|k\rangle$. Using generators allows us to loop only once over time to obtain the time-evolved states `psi_t` and `n_psi_t`. In doing so, we evaluate the expectation value $\langle \mathrm{FS}(t)|n_i(0)|\mathrm{nFS}(t)\rangle$ using the `matrix_ele()` method of the `hamiltonian` class. The flag

[1] The `exp_op` class uses the scipy functions `scipy.linalg.expm()` (when the user requests to compute the matrix exponential explicitly) and `scipy.sparse.linalg.expm_multiply()`. Note also the existence of the QuSpin function `tools.evolution.expm_multiply_parallel()` which provides an OpenMP implementation of `scipy.sparse.linalg.expm_multiply()`, but is not part of the `exp_op` class (when only the application of the matrix exponential on a state is required).

diagonal=True makes sure only the diagonal matrix elements are calculated[2] and returned as a one-dimensional array.

```
# preallocate variable
correlators=np.zeros(t.shape+psi0.shape[1:])
# loop over the time-evolved states
for i, (psi,n_psi) in enumerate( zip(psi_t,n_psi_t) ):
    correlators[i,:]=n_2.matrix_ele(psi,n_psi,diagonal=True).real
```

Finally, we weigh all singe-state correlators by the Fermi-Dirac distribution to obtain the finite-temperature non-equal time correlation function $C_{0,L/2}(t,\beta)$.

```
# evaluate correlator at finite temperature
n_FD=1.0/(np.exp(beta*E)+1.0)
correlator = (n_FD*correlators).sum(axis=-1)
```

The complete code including the lines that produce Fig. 2 is available under:

http://weinbe58.github.io/QuSpin/examples/example5.html

### 2.3 Many-Body Localization in the Fermi-Hubbard Model

This example shows how to:

- construct Hamiltonians for spinful fermions using the spinful_fermion_basis_1d class,
- use quantum_operator class to construct Hamiltonians with varying parameters,
- use new basis.index() function to construct Fock states from strings,
- use obs_vs_time() functionality to measure observables as a function of time.

A class of exciting new problems in the field of non-equalibrium physics is that of many-body localised (MBL) models. The MBL transition is a dynamical phase transition in the eigenstates of a many-body Hamiltonian. Driven primarily by quenched disorder, the transition is distinguished by the system having ergodic eigenstates in the weak disorder limit and non-ergodic eigenstates in the strong disorder limit. The MBL phase is reminiscent of integrable systems as one can construct quasi-local integrals of motion, but these integrals of motion are much more robust in the sense that they are not sensitive to small perturbations, as is the case in many classes of integrable systems [27, 28, 81–83].

Motivated by recent experiments [84–93] we explore MBL in the context of fermions using QuSpin. The model we will consider is the Fermi-Hubbard model with quenched random disorder which has the following Hamiltonian:

$$H = -J \sum_{i=0,\sigma}^{L-2} \left( c^\dagger_{i\sigma} c_{i+1,\sigma} - c_{i\sigma} c^\dagger_{i+1,\sigma} \right) + U \sum_{i=0}^{L-1} n_{i\uparrow} n_{i\downarrow} + \sum_{i=0,\sigma}^{L-1} V_i n_{i\sigma} , \tag{9}$$

where $c_{i\sigma}$ and $c^\dagger_{i\sigma}$ are the fermionic creation and annihilation operators on site $i$ for spin $\sigma$, respectively. We will work in the sector of $1/4$ filling for both up and down spins. The observable of interest is the sublattice imbalance [85, 87, 91]:

$$I = (N_A - N_B)/N_{\rm tot} , \tag{10}$$

[2]Recall that psi_t and n_psi_t contain many time-evolved states, and if one uses the default diagonal=False, all off-diagonal matrix elements will be computed as well, so the result will be a two-dimensional array

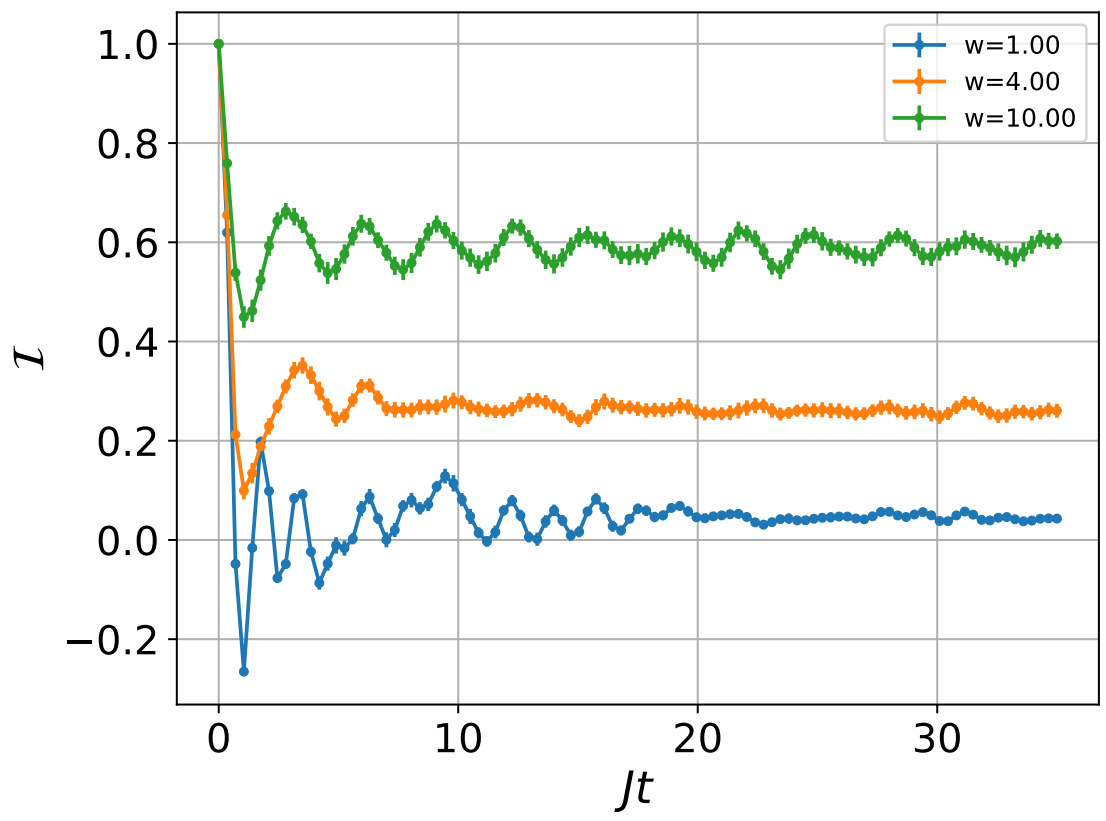

Figure 3: Sublattice imbalance $I$ as a function of time, averaged over 100 disorder realizations for different disorder strengths. This data was taken on a chain of length $L = 8$.

where $A$ and $B$ refer to the different sublattices of the chain and $N$ is the particle number.

We prepare an initial configuration of fermions of alternating spin on every other site. Evolving this initial state under the Hamiltonian (9), we calculate the time dependence of the imbalance $I(t)$ which provides useful information about the ergodicity of the Hamiltonian $H$ (or the lack thereof). If the Hamiltonian is ergodic, the imbalance should decay to zero in the limit $t \to \infty$, as one would expect due to thermalising dynamics. On the other hand, if the Hamiltonian is MBL then some memory of the initial state will be retained at all times, and therefore the imbalance $I(t)$ will remain non-zero even at infinite times.

Because the Hilbert space dimension grows so quickly with the lattice size for the Fermi-Hubbard Hamiltonian, we only consider the dynamics after a finite amount of time, instead of looking at infinite-time expectation values which require knowledge of the entire basis of the Hamiltonian, which requires more computational resources.

*Code Analysis*—We start out by loading a set of libraries which we need to proceed with the simulation of MBL fermions.

```
from quspin.operators import hamiltonian,exp_op,quantum_operator # operators
from quspin.basis import spinful_fermion_basis_1d # Hilbert spaces
from quspin.tools.measurements import obs_vs_time # calculating dynamics
import numpy as np # general math functions
from numpy.random import uniform,choice # tools for doing random sampling
from time import time # tool for calculating computation time
```

While we already encountered most of these libraries and functions, in this example we introduce the `quantum_operator` class which defines an operator, parametrized by multiple parameters, as opposed to the Hamiltonian which is only parametrized by time. Also, since this example requires us to do many different disorder realisations, we use `NumPy`'s random number library to do random sampling. We load `uniform` to generate the uniformly distributed random potential as well as `choice` which we use to estimate the uncertainties of the disorder averages using a bootstrap re-sampling procedure which we explain below. In order to time how long each realization takes, we use the `time` function from python's `time` library.

After importing all the required libraries and functions we set up the parameters for the simulation including the number of realizations `n_real`, and the physical couplings `J`, `U`, the number of up and down fermions, etc.

```
##### setting parameters for simulation
# simulation parameters
```

```
12 n_real = 100 # number of realizations
13 n_boot = 100 # number of bootstrap samples to calculate error
14 # physical parameters
15 L = 8 # system size
16 N = L//2 # number of particles
17 N_up = N//2 + N % 2 # number of fermions with spin up
18 N_down = N//2 # number of fermions with spin down
19 w_list = [1.0,4.0,10.0] # disorder strength
20 J = 1.0 # hopping strength
21 U = 5.0 # interaction strength
22 # range in time to evolve system
23 start,stop,num=0.0,35.0,101
24 t = np.linspace(start,stop,num=num,endpoint=True)
```

Next we set up the basis, introducing here the `spinful_fermion_basis_1d` constructor class. It works the usual way, except the particle number sector `N_f` is now a tuple containing the number of up and down fermions.

```
26 ###### create the basis
27 # build spinful fermions basis
28 basis = spinful_fermion_basis_1d(L,Nf=(N_up,N_down))
```

The next step in the procedure is to set up the site-coupling and operator lists:

```
30 ##### create model
31 # define site-coupling lists
32 hop_right = [[-J,i,i+1] for i in range(L-1)] # hopping to the right OBC
33 hop_left = [[J,i,i+1] for i in range(L-1)] # hopping to the left OBC
34 int_list = [[U,i,i] for i in range(L)] # onsite interaction
35 # site-coupling list to create the sublattice imbalance observable
36 sublat_list = [[(-1.0)**i/N,i] for i in range(0,L)]
37 # create static lists
38 operator_list_0 = [
39             ["+-|", hop_left], # up hop left
40             ["-+|", hop_right], # up hop right
41             ["|+-", hop_left], # down hop left
42             ["|-+", hop_right], # down hop right
43             ["n|n", int_list], # onsite interaction
44             ]
45 imbalance_list = [["n|",sublat_list],["|n",sublat_list]]
```

Notice here that the `"|"` character is used to separate the operators which belong to the up (left side of tensor product) and down (right side of tensor product) Hilbert spaces in `basis`. If no string in present the operator is assumed to be the identity `'I'`. The site-coupling list, on the other hand, does not require separating the two sides of the tensor product, as it is assumed that the operator string lines up with the correct site index when the `'|'` character is removed.

In the last couple of lines defining model (see below), we create a python `dictionary` object in which we add static operator lists as `values`, indexed by a particular string known as the corresponding `key`. This dictionary, which we refer to as `operator_dict`, is then passed into the `quantum_operator` class which, for each `key` in `operator_dict`, constructs the operator listed in the site-coupling list for that `key`. When one wants to evaluate this operator for a particular set of parameters, one uses a second dictionary (see `params_dict` below in lines 74 − 76) with the same keys as `operator_dict`: the value corresponding to each key in the `params_dict` multiples the operator stored at that same key in `operator_dict`. This allows one to parametrize many-body operators in more complicated and general ways. In the present example, we define a key for the Fermi-Hubbard Hamiltonian, and then, as we need to

change the disorder strength in between realisations, we also define keys for the local density operator for both up and down spins on each site. By doing this, we can then construct any disordered Hamiltonian by specify the disorder at each site, and changing its value from one realisation to another.

```
# create operator dictionary for quantum_operator class
# add key for Hubbard hamiltonian
operator_dict=dict(H0=operator_list_0)
# add keys for local potential in each site
for i in range(L):
    # add to dictioanry keys h0,h1,h2,...,hL with local potential operator
    operator_dict["n"+str(i)] = [["n|",[[1.0,i]]],["|n",[[1.0,i]]]]
```

The `quantum_operator` class constructs operators in almost an identical manner as a `hamiltonian()` class with the exception that there is no dynamic list. Next, we construct our initial state with the fermions dispersed over the lattice on every other site. To get the index of the `basis` state which this initial state corresponds to, one can use the `index` function of the `spinful_fermion_basis_1d` class. This function takes a string or integer representing the product state for each of the Hilbert spaces and then searches to find the full product state, returning the corresponding index. We can then create an empty array `psi_0` of dimension the total Hilbert space size, and insert unity at the index corresponding to the initial state. This allows us to easily define the many-body product state used in the rest of the simulation.

```
###### setting up operators
# set up hamiltonian dictionary and observable (imbalance I)
no_checks = dict(check_pcon=False,check_symm=False,check_herm=False)
H_dict = quantum_operator(operator_dict,basis=basis,**no_checks)
I = hamiltonian(imbalance_list,[],basis=basis,**no_checks)
# strings which represent the initial state
s_up = "".join("1000" for i in range(N_up))
s_down = "".join("0010" for i in range(N_down))
# basis.index accepts strings and returns the index
# which corresponds to that state in the basis list
i_0 = basis.index(s_up,s_down) # find index of product state
psi_0 = np.zeros(basis.Ns) # allocate space for state
psi_0[i_0] = 1.0 # set MB state to be the given product state
print("H-space size: {:d}, initial state: |{:s}>(x)|{:s}>".format(basis.Ns,s_up,
    s_down))
```

Now that the operators are all set up, we can proceed with the simulation of the dynamics. First, we define a function which, given a disorder realization of the local potential, calculates the time evolution of $\mathcal{I}(t)$. We shall guide the reader through this function step by step. The syntax for this begins as follows:

```
# define function to do dynamics for different disorder realizations.
def real(H_dict,I,psi_0,w,t,i):
    # body of function goes below
    ti = time() # start timing function for duration of reach realisation
```

The first step is to construct the Hamiltonian from the `disorder` list which is as simple as

```
    # create a parameter list which specifies the onsite potential with disorder
    params_dict=dict(H0=1.0)
    for j in range(L):
        params_dict["n"+str(j)] = uniform(-w,w)
    # using the parameters dictionary construct a hamiltonian object with those
    # parameters defined in the list
    H = H_dict.tohamiltonian(params_dict)
```

using the tohamiltonian() method of H_dict class, which accepts as argument the dictionary params_dict, which shares the same keys as operator_dict, but whose values are determined by the disorder list which changes from one realisation to another.

Once the Hamiltonian has been constructed we want to time-evolve the initial state with it. To this end, we use the fact that, for time-independent Hamiltonians, the time-evolution operator coincides with the matrix exponential $\exp(-itH)$. In QuSpin, a convenient way to define matrix exponentials is offered by the exp_op class. Given an operator $A$ , it calculates $\exp(atA)$ for any complex-valued number $a$. The time grid for $t$ is specified using the optional arguments start, stop and num. If the latter are omitted, default is $t = 1$. The exp_op objects are either a list of arrays, the elements of which correspond to the operator $\exp(atA)$ at the times defined or, if iterate=True a generator for this list is returned. Here, we use exp_op to create a generator list containing the time-evolved states as follows

```
    # use exp_op to get the evolution operator
    U = exp_op(H,a=-1j,start=t.min(),stop=t.max(),num=len(t),iterate=True)
    psi_t = U.dot(psi_0) # get generator psi_t for time evolved state
```

To calculate the expectation value of the imbalance operator in time, we use the obs_vs_time function. Since the usage of this function was extensively discussed in Example 2 and Example 3 of Ref. [16], here we only mention that it accepts a (collection of) state(s) [or a generator] psi_t, a time vector t, and a dictionary dict(I=I), whose values are the observables to calculate the expectation value of. The function returns a dictionary, the keys of which correspond to the keys every observable was defined under.

```
    # use obs_vs_time to evaluate the dynamics
    t = U.grid # extract time grid stored in U, and defined in exp_op
    obs_t = obs_vs_time(psi_t,t,dict(I=I))
```

The function ends by printing the time of executing and returning the value for $I$ as a function of time for this realization

```
    # print reporting the computation time for realization
    print("realization {}/{} completed in {:.2f} s".format(i+1,n_real,time()-ti))
    # return observable values
    return obs_t["I"]
```

Now we are all set to run the disorder realizations for the different disorder strengths which in principle can be spilt up over multiple simulations, e.g. using joblib [c.f. Example 1 from Ref. [16]] but for completeness we do all of the calculations in one script.

```
###### looping over differnt disorder strengths
for w in w_list:
    I_data = np.vstack([real(H_dict,I,psi_0,w,t,i) for i in range(n_real)])
```

Last, we calculate the average and its error bars using *bootstrap re-sampling*. The idea is that we have a given set of n_real samples from which we can select randomly a set of n_real samples with replacements. This means that sometimes we will get the same individual sample represented multiple times in one of these sets. Then by averaging over this set one can get an estimate of the mean value for the original sample set. This mean value is what is called a bootstrap sample. In principle there are a very large number of possible bootstrap samples so in practice one only takes a small fraction of the total set of bootstrap samples from which you can estimate things like the variance of the mean which is the error of the original sample set.

```
    ##### averaging and error estimation
    I_avg = I_data.mean(axis=0) # get mean value of I for all time points
    # generate bootstrap samples
    bootstrap_gen = (I_data[choice(n_real,size=n_real)].mean(axis=0) for i in
    range(n_boot))
```

```
    # generate the fluctuations about the mean of I
    sq_fluc_gen = ((bootstrap-I_avg)**2 for bootstrap in bootstrap_gen)
    I_error = np.sqrt(sum(sq_fluc_gen)/n_boot)
```

The complete code including the lines that produce Fig. 3 is available under:

http://weinbe58.github.io/QuSpin/examples/example6.html

### 2.4 The Bose-Hubbard Model on Translationally Invariant Ladder

This example shows how to:

- construct interacting Hamiltonians for bosonic systems,
- construct a Hamiltonian on a ladder geometry,
- use the `block_ops` class to evolve a state over several symmetry sectors at once,
- measure the entanglement entropy of a state on the ladder.

*Physics Setup*— In this example we use QuSpin to simulate the dynamics of the Bose-Hubbard model (BHM) on a ladder geometry. The BHM is a minimal model for interacting lattice bosons [74] which is most often experimentally realizable in cold atom experiments [94]. The Hamiltonian is given by

$$H_{\mathrm{BHM}} = -J\sum_{\langle ij\rangle}\left(b_i^\dagger b_j + \mathrm{h.c.}\right) + \frac{U}{2}\sum_i n_i(n_i-1), \tag{11}$$

where $b_i$ and $b_i^\dagger$ are bosonic creation and annihilation operators on site $i$, respectively, and $\langle ij\rangle$ denotes nearest neighbors on the ladder. We consider a half-filled ladder of length $L$ with $N=2L$ sites and cylindrical boundary condition, i.e. a periodic boundary condition along the ladder-leg direction. We are interested in the strongly-interacting limit $U/J \gg 1$, where the mean-field Gross-Pitaevskii theory fails. Hence, we restrict the local Hilbert space to allow at most two particles per site, effectively using a total of three states per site: empty, singly and doubly occupied.

The system is initialised in a random Fock state, and then evolved with the Hamiltonian (11). Since the BHM is non-integrable, we expect that the system eventually thermalizes, so that the long-time occupation becomes roughly uniform over the entire system. Besides the local density, we also measure the entanglement entropy between the two legs of the ladder.

As we consider a translational invariant ladder, the Hilbert space factorizes into subspaces corresponding to the different many-body momentum blocks. This is similar to what was discussed in Sec. 2.2 for the SSH model, but slightly different as we consider translations of the *many-body* Fock states as opposed to the single particle states [95]. In certain cases, it happens that, even though the Hamiltonian features symmetries, the initial state does not obey them, as is the case in the present example, where the initial state is a random Fock state. Thus, in this section we project the wavefunction to the different symmetry sectors and evolve each of the projections separately under the Hamiltonian for that symmetry sector. After the evolution, each of these symmetry-block wavefunctions is transformed back to the local Fock space basis, and summed up to recover the properly evolved initial state. We can then measure quantities such as the on-site density and the entanglement entropy. Figure 4(a) show the entanglement density between the two legs as a function of time, while Figure 4(b) shows a heat map of the local density of the bosonic gas. Both quantities show that after a short period of time the gas is completely thermalized.

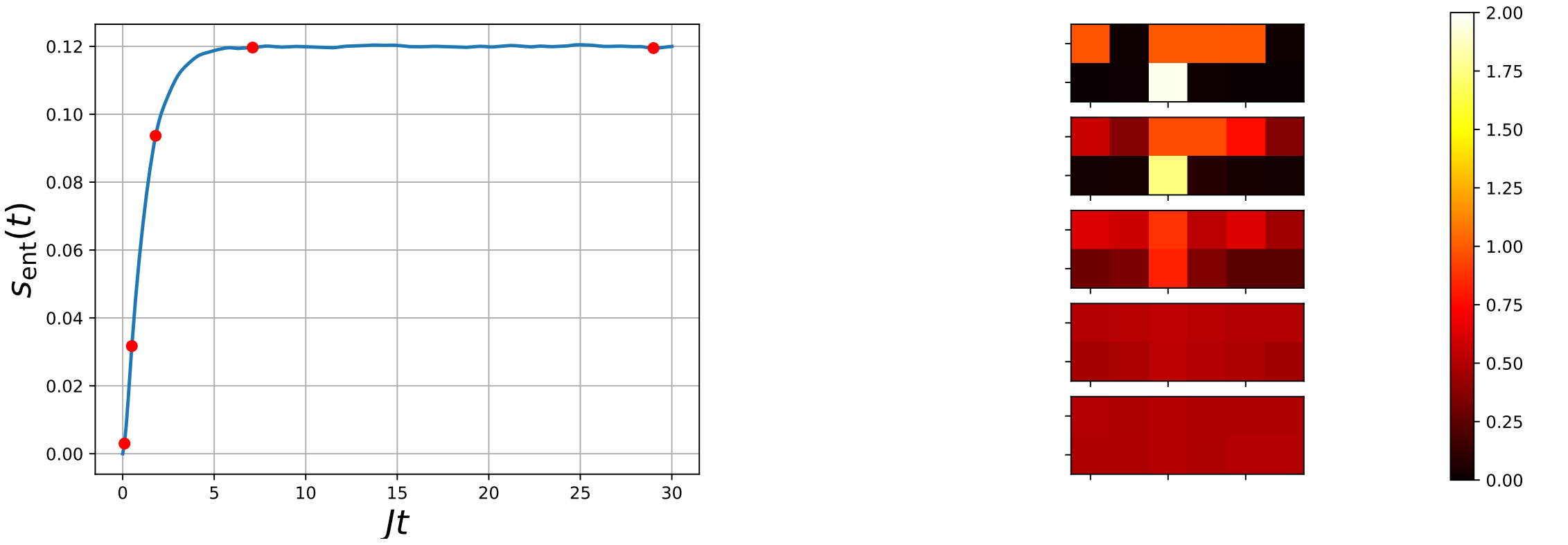

(a) Entanglement entropy density between the two legs of the ladder as a function of time.

(b) Local density of bosons as a function of time. Later times appearing farther down the page.

Figure 4: Quantities measured in the BHM: (a) the leg-to-leg entanglement entropy density and (b) the local density on each site as a function of time. The red dots in (a) show the time points at which the snapshots of the local density (b) are taken. The parameters are $U/J = 20$, and $L = 6$.

*Code Analysis*—Let us now show how one can simulate this system using QuSpin. Following the code structure of previous examples, we begin by loading the modules needed for the computation:

```
from __future__ import print_function, division #import python 3 functions
from quspin.operators import hamiltonian # Hamiltonians and operators
from quspin.basis import boson_basis_1d # bosonic Hilbert space
from quspin.tools.block_tools import block_ops # dynamics in symmetry blocks
import numpy as np # general math functions
import matplotlib.pyplot as plt # plotting library
import matplotlib.animation as animation # animating movie of dynamics
```

First, we set up the model parameters defining the length of the ladder L, the total number of sites in the chain N=2*L, as well as the filling factor for the bosons nb, and the maximum number of states per site (i.e. the local on-site Hilbert space dimension) sps. The hopping matrix elements $J_\perp$, $J_{||,1}$, and $J_{||,2}$ (see Code Snippet 1) correspond to the python script variables J_perp, J_par_1, and J_par_2, respectively. The on-site interaction is denoted by U.

```
##### define model parameters
# initial seed for random number generator
np.random.seed(0) # seed is 0 to produce plots from QuSpin2 paper
# setting up parameters of simulation
L = 6 # length of chain
N = 2*L # number of sites
nb = 0.5 # density of bosons
sps = 3 # number of states per site
J_par_1 = 1.0 # top side of ladder hopping
J_par_2 = 1.0 # bottom side of ladder hopping
J_perp =  0.5 # rung hopping
U = 20.0 # Hubbard interaction
```

Let us proceed to construct the Hamiltonian and the observables for the problem. For bosonic systems, we have '+','-' 'n', and 'I' as available Fock space operators to use. In order to set up the local Hubbard interaction, we first split it up in two terms: $U/2\sum_i n_i(n_i-1) = -U/2\sum_i n_i + U/2\sum_i n_i^2$. Thus, we need two coupling lists:

```
##### set up Hamiltonian and observables
# define site-coupling lists
int_list_1 = [[-0.5*U,i] for i in range(N)] # interaction $-U/2 \sum_i n_i$
int_list_2 = [[0.5*U,i,i] for i in range(N)] # interaction: $U/2 \num_i n_i^2$
```

*Code Snippet* 1: Translationally Invariant Ladder Graph and its Symemtries

```
###### ladder lattice
# hopping coupling parameters:
# - : J_par_1
# = : J_par_2
# | : J_perp
#
# lattice graph
#
 = 1 = 3 = 5 = 7 = 9 =
   |   |   |   |   |
 - 0 - 2 - 4 - 6 - 8 -
#
# translations along leg-direction (i -> i+2):
#
 = 9 = 1 = 3 = 5 = 7 =
   |   |   |   |   |
 - 8 - 0 - 2 - 4 - 6 -
#
# if J_par_1=J_par_2, one can use regular chain parity (i -> N - i) as
    combination
# of the two ladder parity operators:
#
 - 8 - 6 - 4 - 2 - 0 -
   |   |   |   |   |
 - 9 - 7 - 5 - 3 - 1 -
```

We also define the hopping site-coupling list. In general, QuSpin can set up the Hamiltonian on any graph (thus, including higher-dimensions). By labelling the lattice sites conveniently, we can define the ladder geometry, as shown in Code Snippet 1. For the current ladder geometry of interest, there are three types of hopping: one along each of the two legs with tunnelling matrix elements `J_par_1` and `J_par_1`, respectively, and the transverse hopping along the rungs of the ladder with strength `J_perp`. The even sites correspond to the bottom leg while the odds sites are the top leg of the ladder. Therefore, the hopping on the bottom/top leg is defined by `[J_par_...,i,(i+2)%N]`, while the rung hopping (from the top leg to the bottom leg) is defined by `[J_perp,i,(i+1)%N]`.

```
# setting up hopping lists
hop_list = [[-J_par_1,i,(i+2)%N] for i in range(0,N,2)] # PBC bottom leg
hop_list.extend([[-J_par_2,i,(i+2)%N] for i in range(1,N,2)]) # PBC top leg
hop_list.extend([[-J_perp,i,i+1] for i in range(0,N,2)]) # perp/rung hopping
hop_list_hc = [[J.conjugate(),i,j] for J,i,j in hop_list] # add h.c. terms
```

where we used the `list_1.extend(list_2)` method to concatenate two lists together[3].

Next, we define the static and dynamic lists, which are needed to construct the Hamiltonian.

```
# set up static and dynamic lists
static = [
```

[3] Note that the `extend` function is done inplace so if one tries to do `new_list=list_1.extend(list_2)`, `new_list` will be `None` and `list_1` will have all of the elements of `list_2` appended to it.

```
            ["+-",hop_list], # hopping
            ["-+",hop_list_hc], # hopping h.c.
            ["nn",int_list_2], # U n_i^2
            ["n",int_list_1] # -U n_i
        ]
dynamic = [] # no dynamic operators
```

Instead of creating a `hamiltonian` class object in real space, we use the `block_ops` class to set up the Hamiltonian in a block-diagonal form in momentum space, similar to the SSH model, c.f. Sec. 2.2, but now genuinely many-body. In order to reduce the computational cost, the state is evolved in momentum space and projected to the Fock basis after the calculation. The purpose of `block_ops` is to provide a simple interface for solving the Schrödinger equation when an initial state does not obey the symmetries of the Hamiltonian it is evolved under. We have seen an example of this in Sec. 2.2 when trying to measure non-equal space-time correlation functions of local operators in a translational invariant system, while in this section we explicitly start out with a state which does not obey translation invariance. To construct the `block_ops` object we use the follow set of lines, explained below:

```
# create block_ops object
blocks=[dict(kblock=kblock) for kblock in range(L)] # blocks to project on to
baisis_args = (N,) # boson_basis_1d manditory arguments
basis_kwargs = dict(nb=nb,sps=sps,a=2) # boson_basis_1d optional args
get_proj_kwargs = dict(pcon=True) # set projection to full particle basis
H_block = block_ops(blocks,static,dynamic,boson_basis_1d,baisis_args,np.
    complex128,
                    basis_kwargs=basis_kwargs,get_proj_kwargs=get_proj_kwargs)
```

First, we create a list of dictionaries `blocks` which defines the different symmetry sectors to project the initial state to, before doing the time evolution[4]. The optional arguments `basis_args` and `basis_kwargs` apply to every symmetry sector. Last, `get_proj_kwargs` contains the optional arguments to construct the projectors[5]. This class automatically projects the initial state onto the different symmetry sectors and will evolve the sectors individually in serial or with an additional option in parallel over multiple cpu cores. After the evolution the class reconstructs the state and returns it back to the user. The Hamiltonian and the evolution itself will only be calculated for the sectors which have non-vanishing overlap with the initial state. Furthermore, the class will, by default, construct things on the fly as it needs them to save the user memory. For more information about this class we refer the user to the Documentation, c.f. App. C.

Finally, we define the local density operators on the full particle-conserving Hilbert space using the `boson_basis_1d` class.

```
# setting up local Fock basis
basis = boson_basis_1d(N,nb=nb,sps=sps)
# setting up observables
no_checks = dict(check_herm=False,check_symm=False,check_pcon=False)
n_list = [hamiltonian([["n",[[1.0,i]]]],[],basis=basis,dtype=np.float64,**
    no_checks) for i in range(N)]
```

Having set up the Hamiltonian, we now proceed to the time-evolution part of the problem. We begin by defining the initial random state in the Fock basis.

```
# set up initial state
i0 = np.random.randint(basis.Ns) # pick random state from basis set
```

[4] `block_ops` will not evolve in those symmetry sectors for which the projection is 0.

[5] In this case setting `pcon=True` means that the projector takes the state *from* the symmetry reduced basis *to* the fixed particle number basis.

```
psi = np.zeros(basis.Ns,dtype=np.float64)
psi[i0] = 1.0
# print info about setup
state_str = "".join(str(int((basis[i0]//basis.sps**(L-i-1)))%basis.sps) for i in range(N))
print("total H-space size: {}, initial state: |{}>".format(basis.Ns,state_str))
```

Next we define the times which we would like to solve the Schrödinger equation for. Since the Hamiltonian is time-independent, we use the `exp_op` class to compute the time-evolution operator as the exponential of the Hamiltonian, see Sec. 2.3. Therefore, we consider linearly spaced time points defined by the variables `start`,`stop`, and `num`.

```
# setting up parameters for evolution
start,stop,num = 0,30,301 # 0.1 equally spaced points
times = np.linspace(start,stop,num)
```

To calculate the states as a function of time, we use the `expm` function of the `block_ops` class to first construct the unitary evolution operator as the matrix exponential of the time-independent Hamiltonian[6]. We define the `expm` function to have almost identical arguments as that of the `exp_op` class, but with some major exceptions. For one, because the Hamiltonian factorizes in a block-diagonal form, the evolution over each block can be done separately (e.g. just trivially loop through the blocks sequentially). In some cases, however, e.g. for single particle Hamiltonians [see Sec. 2.2], there are a lot of small blocks, and it actually makes sense to calculate the matrix exponential in block diagonal form which is achieved by setting the optional argument `block_diag=True`. Another optional argument, `n_jobs=int`, allows the user to spawn multiple python processes which do the calculations for the different blocks simultaneously for `n_jobs>1`. On most systems these processes will be distributed over multiple CPUs which can speed up the calculations if there are resources for this available. This also works in conjunction with the `block_diag` flag where each process creates its own block diagonal matrix for the calculation. Once all the calculations for each block are completed, the results are combined and conveniently projected back to the original local Fock basis automatically.

```
# calculating the evolved states
n_jobs = 1 # paralelisation: increase to see if calculation runs faster!
psi_t = H_block.expm(psi,a=-1j,start=start,stop=stop,num=num,block_diag=False,n_jobs=n_jobs)
```

We can now use the time dependent states calculated to compute the expectation value of the local density

```
# calculating the local densities as a function of time
expt_n_t = np.vstack([n.expt_value(psi_t).real for n in n_list]).T
# reshape data for plotting
n_t = np.zeros((num,2,L))
n_t[:,0,:] = expt_n_t[:,0::2]
n_t[:,1,:] = expt_n_t[:,1::2]
```

We can also compute the entanglement entropy between the two legs of the ladder. In the newer versions of QuSpin we have moved the entanglement entropy calculations to the `basis` classes themselves (keeping of course backwards compatible functions from older versions). Once again, we refer the reader to the Documentation to learn more about how to use this function, see App. C.

```
# calculating entanglement entropy
sub_sys_A = range(0,N,2) # bottom side of ladder
gen = (basis.ent_entropy(psi,sub_sys_A=sub_sys_A)["Sent_A"]/L for psi in psi_t.T[:])
```

[6]For time dependent Hamiltonians, the `block_ops` class contains a method called `evolve`, see App. C.

```
ent_t = np.fromiter(gen,dtype=np.float64,count=num)
```

The complete code including the lines that produce Fig. 4 is available under:

http://weinbe58.github.io/QuSpin/examples/example7.html

### 2.5 The Gross-Pitaevskii Equation and Nonlinear Time Evolution

This example shows how to:

- simulate user-defined time-dependent nonlinear equations of motion using the `evolution.evolve()` routine,
- use imaginary time dynamics to find the lowest energy state.

*Physics Setup*—The Gross-Pitaevskii wave equation (GPE) has been shown to govern the physics of weakly-interacting bosonic systems. It constitutes the starting point for studying Bose-Einstein condensates [96], but can also appear in non-linear optics [97], and represents the natural description of Hamiltonian mechanics in the wave picture [98]. One of its interesting features is that it can exhibit chaotic classical dynamics, a physical manifestation of the presence of a cubic non-linear term.

Here, we study the time-dependent GPE on a one-dimensional lattice:

$$\begin{aligned} i\partial_t\psi_j(t) &= -J\left(\psi_{j-1}(t)+\psi_{j+1}(t)\right)+\frac{1}{2}\kappa_{\text{trap}}(t)(j-j_0)^2\psi_j(t)+U|\psi_j(t)|^2\psi_j(t)\,,\\ \kappa_{\text{trap}}(t) &= (\kappa_f-\kappa_i)t/t_{\text{ramp}}+\kappa_i\,, \end{aligned} \tag{12}$$

where $J$ is the hopping matrix element, $\kappa_{\text{trap}}(t)$ – the harmonic trap strength which varies slowly in time over a scale $t_{\text{ramp}}$, and $U$ – the mean-field interaction strength. The lattice sites are labelled by $j=0,\dots,L-1$, and $j_0$ is the centre of the 1$d$ chain. We set the lattice constant to unity, and use open boundary conditions.

Whenever $U=0$, the system is non-interacting and the GPE reduces to the Heisenberg EOM for the bosonic field operator $\hat{\psi}_j(t)$. Thus, for the purposes of using QuSpin to simulate the GPE, it is instructive to cast Eq. (12) in the following generic form

$$i\partial_t\vec{\psi}(t)=H_{\text{sp}}(t)\vec{\psi}(t)+U\vec{\psi}^*(t)\circ\vec{\psi}(t)\circ\vec{\psi}(t), \tag{13}$$

where $[\vec{\psi}(t)]_j=\psi_j(t)$, and $\circ$ represents the element-wise multiplication

$$\vec{\psi}(t)\circ\vec{\phi}(t)=\Big(\psi_0(t)\phi_0(t),\psi_1(t)\phi_1(t),\dots,\psi_{L-1}(t)\phi_{L-1}(t)\Big)^t.$$

The time-dependent single-particle Hamiltonian in real space is represented as an $L\times L$ matrix, $H_{\text{sp}}(t)$, which comprises the hopping term, and the harmonic trap.

We want to initiate the time-evolution of the system at $t=0$ in its lowest energy state. To this end, we can define a 'ground state' for the GPE equation, in terms of the configuration which minimises the energy of the system:

$$\begin{aligned} \vec{\psi}_{\text{GS}} &= \inf_{\vec{\psi}}\left(\vec{\psi}^tH_{\text{sp}}(0)\vec{\psi}+\frac{U}{2}\sum_{j=0}^{L-1}|\psi_j|^4\right)\\ &= \inf_{\psi_j}\left(-J\sum_{j=0}^{L-2}(\psi^*_{j+1}\psi_j+\text{c.c.})+\frac{1}{2}\kappa_{\text{trap}}(0)\sum_{j=0}^{L-1}(j-j_0)^2|\psi_j|^2+\frac{U}{2}\sum_{j=0}^{L-1}|\psi_j|^4\right). \end{aligned} \tag{14}$$

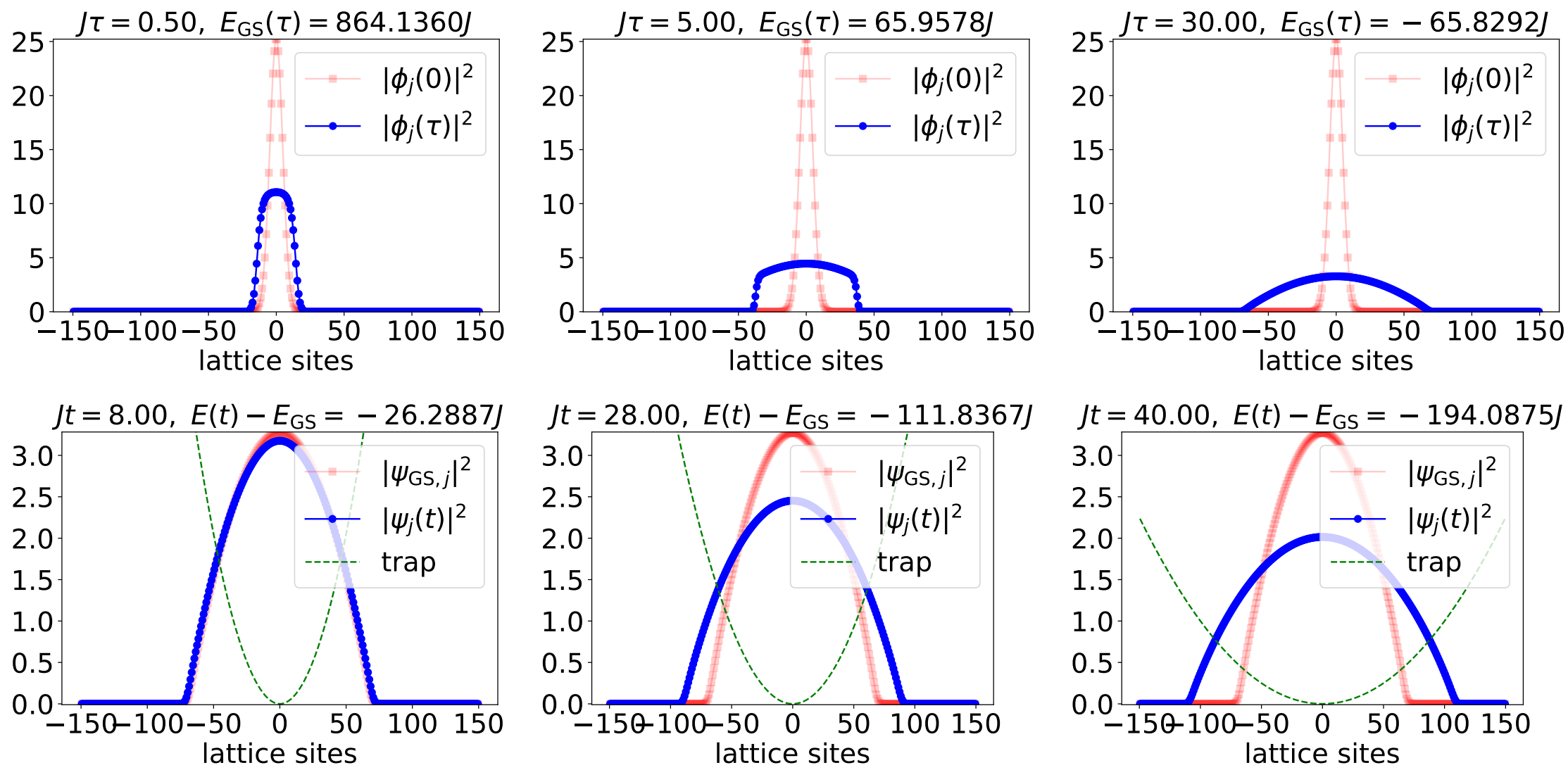

Figure 5: Top row: three snapshots of the imaginary time dynamics which turns the non-interacting ground state into the GPE ground state at long imaginary times. Bottom row: three snapshots of the real-time evolution induced by slowly widening the harmonic trap strength. The pale red curve is the initial state, while the blue curve shows the instantaneous state. The harmonic trap is shown with a dashed green line.

One way to find the configuration $\vec{\psi}_{\rm GS}$, is to solve the GPE in imaginary time ($it \to \tau$), which induces exponential decay in all modes of the system, and singles out the lowest-energy state in the longer run. In doing so, we keep the norm of the solution fixed:

$$\begin{aligned} \partial_\tau \vec{\varphi}(\tau) &= -\Big[ H_{\rm sp}(0)\vec{\varphi}(\tau) + U\vec{\varphi}^*(\tau)\circ\vec{\varphi}(\tau)\circ\vec{\varphi}(\tau)\Big], \qquad ||\vec{\varphi}(\tau)|| = \text{const.}, \\ \vec{\psi}_{\rm GS} &= \lim_{\tau\to\infty} \vec{\varphi}(\tau). \end{aligned} \tag{15}$$

Snapshots of the imaginary-time evolution can be seen in Fig. (5), top row.

Once we have the initial state $\vec{\psi}_{\rm GS}$, we evolve it according to the time-dependent GPE, Eq. (12), and track down the time evolution of the condensate density $\rho_j(t) = |\psi_j(t)|^2$. Fig. 5 (bottom row) shows snapshots of the state as it evolves.

*Code Analysis*—In the following, we demonstrate how one can code the above physics using QuSpin. As usual, we begin by loading the necessary packages:

```
from __future__ import print_function, division
from quspin.operators import hamiltonian # Hamiltonians and operators
from quspin.basis import boson_basis_1d # Hilbert space boson basis
from quspin.tools.evolution import evolve # nonlinear evolution
import numpy as np # generic math functions
import matplotlib.pyplot as plt # plotting library
from six import iteritems # loop over elements of dictionary
```

Next, we define the model parameters. We distinguish between static parameters and dynamic parameters – those involved in the time-dependent trap widening.

```
##### define model parameters #####
L=300 # system size
# calculate centre of chain
if L%2==0:
```

```
    j0 = L//2-0.5 # centre of chain
else:
    j0 = L//2 # centre of chain
sites=np.arange(L)-j0
# static parameters
J=1.0 # hopping
U=1.0 # Bose-Hubbard interaction strength
# dynamic parameters
kappa_trap_i=0.001 # initial chemical potential
kappa_trap_f=0.0001 # final chemical potential
t_ramp=40.0/J # set total ramp time
```

In order to do time evolution, we code up the trap widening protocol from Eq. (12) in the function `ramp`. Since we want to make use of QuSpin's time-dependent operator features, the first argument must be the time `t`, followed by all protocol parameters. These same parameters are then explicitly stored in the variable `ramp_args`.

```
# ramp protocol
def ramp(t,kappa_trap_i,kappa_trap_f,t_ramp):
    return  (kappa_trap_f - kappa_trap_i)*t/t_ramp + kappa_trap_i
# ramp protocol parameters
ramp_args=[kappa_trap_i,kappa_trap_f,t_ramp]
```

With this, we are ready to construct the single-particle Hamiltonian $H_{\mathrm{sp}}(t)$. The first step is to define the site-coupling lists, and the static and dynamic lists. Note that the dynamic list, which defines the harmonic potential in the single-particle Hamiltonian, contains four elements: apart from the operator string and the corresponding site-coupling list, the third and fourth elements are the time-dependent function `ramp` and its argument list `ramp_args`. We emphasize that this order of appearance is crucial.

```
##### construct single-particle Hamiltonian #####
# define site-coupling lists
hopping=[[-J,i,(i+1)%L] for i in range(L-1)]
trap=[[0.5*(i-j0)**2,i] for i in range(L)]
# define static and dynamic lists
static=[["+-",hopping],["-+",hopping]]
dynamic=[['n',trap,ramp,ramp_args]]
```

To create the single-particle Hamiltonian, we choose to use the bosonic basis constructor `boson_basis_1d` specifying the number sector to `Nb=1` boson for the entire lattice, and a local Hilbert space of `sps=2` states per site (empty and filled).

```
# define basis
basis = boson_basis_1d(L,Nb=1,sps=2)
```

Then we call the `hamiltonian` constructor to build the single-particle matrix. We can obtain the single-particle ground state without fully diagonalising this matrix, by using the sparse diagonalisation method `Hsp.eigsh()`. The `eigsh()` routine accepts the optional flags `k=1` and `which='SA'` which restrict the underlying Lanczos routine to find the first eigenstate starting from the bottom of the spectrum, i.e. the ground state.

```
# build Hamiltonian
Hsp=hamiltonian(static,dynamic,basis=basis,dtype=np.float64)
E,V=Hsp.eigsh(time=0.0,k=1,which='SA')
```

Having set up the Hamiltonian, the next step in the simulation is to compute the ground state of the GPE using imaginary time evolution, c.f. Eq. (15). To this end, we first define the function `GPE_imag_time` which evaluates the RHS. It is required that the first argument for this function is (imaginary) time `tau`, followed by the state `phi`. All other arguments,

such as the single-particle Hamiltonian and the interaction strength are listed last. Note that we evaluate the time-dependent Hamiltonian at `time=0`, since we are interested in finding the GPE ground state for the initial trap configuration $\kappa_i$. Similar to before, we store these optional arguments in a list which we call `GPE_params`.

```
##### imaginary-time evolution to compute GS of GPE #####
def GPE_imag_time(tau,phi,Hsp,U):
    """
    This function solves the real-valued GPE in imaginary time:
    $$ -\dot\phi(\tau) = Hsp(t=0)\phi(\tau) + U |\phi(\tau)|^2 \phi(\tau) $$
    """
    return -( Hsp.dot(phi,time=0) + U*np.abs(phi)**2*phi )
# define ODE parameters
GPE_params = (Hsp,U)
```

Any initial value problem requires us to pick an initial state. In the case of imaginary evolution, this state can often be arbitrary, but needs to be in the same symmetry-sector as the true GPE ground state. Here, we choose the ground state of the single-particle Hamiltonian for an initial state, and normalise it to one particle per site. We also define the imaginary time vector `tau`. This array has to contain sufficiently long times so that we make sure we end up in the long imaginary time limit $\tau \to \infty$, as required by Eq. (15). Since imaginary time evolution is not unitary, QuSpin will be normalising the vector every $\tau$-step. Thus, one also needs to make sure these steps are small enough to avoid convergence problems with the ODE solver.

```
# define initial state to flow to GS from
phi0=V[:,0]*np.sqrt(L) # initial state normalised to 1 particle per site
# define imaginary time vector
tau=np.linspace(0.0,35.0,71)
```

Performing imaginary time evolution is done using the `evolve()` method of the `evolution` submodule. This function accepts an initial state `phi0`, initial time `tau[0]`, and a time vector `tau` and solves any user-defined ODE, here `GPE_imag_time`. The parameters of the ODE are passed using the keyword argument `f_params=GPE_params`. To ensure the normalisation of the state at each $\tau$-step we use the flag `imag_time=True`. Real-valued output can be specified by `real=True`. Last, we request `evolve()` to create a generator object using the keyword argument `iterate=True`. Many of the keyword arguments of `evolve()` are the same as in the `H.evolve()` method of the `hamiltonian` class: for instance, one can choose a specific SciPy solver and pass its arguments, or the solver's absolute and relative tolerance. We refer the interested reader to the documentation, cf. App. C.

```
# evolve state in imaginary time
psi_tau = evolve(phi0,tau[0],tau,GPE_imag_time,f_params=GPE_params,
                            imag_time=True,real=True,iterate=True)
```

Looping over the generator `phi_tau` we have access to the solution, which we display in a form of sequential snapshots:

```
# display state evolution
for i,psi0 in enumerate(psi_tau):
    # compute energy
    E_GS=(Hsp.matrix_ele(psi0,psi0,time=0) + 0.5*U*np.sum(np.abs(psi0)**4) ).real
    # plot wave function
    plt.plot(sites, abs(phi0)**2, color='r',marker='s',alpha=0.2,
                                    label='$|\\phi_j(0)|^2$')
    plt.plot(sites, abs(psi0)**2, color='b',marker='o',
                            label='$|\\phi_j(\\tau)|^2$' )
    plt.xlabel('$\\mathrm{lattice\\ sites}$',fontsize=18)
    plt.title('$J\\tau=%0.2f,\\ E_\\mathrm{GS}(\\tau)=%0.4fJ$'%(tau[i],E_GS)
```

```
                                                                        ,fontsize=18)
    plt.ylim([-0.01,max(abs(phi0)**2)+0.01])
    plt.legend(fontsize=18)
    plt.draw() # draw frame
    plt.pause(0.005) # pause frame
    plt.clf() # clear figure
plt.close()
```

Last, we use our GPE ground state, to time-evolve it in *real* time according to the trap widening protocol `ramp`, hard-coded into the single-particle Hamiltonian. We proceed analogously – first we define the real-time GPE and the time vector. In defining the GPE function, we split the ODE into a time-independent static part and a time-dependent dynamic part. The single-particle Hamiltonian for the former is accessed using the `hamiltonian` attribute `Hsp.static` which returns a SciPy sparse array. We can then manually add the non-linear cubic mean-field interaction term. In order to access the time-dependent part of the Hamiltonian, and evaluate it, we loop over the attribute `Hsp.dynamic`, which is a dictionary, whose keys `fun()` are time-dependent function objects already evaluated at the passed parameter values (here `fun(t)=ramp(t,ramp_args)`). These functions `fun()` accept only one argument: the time value. The value corresponding to the key of the `H_sp.dynamic` dictionary is the matrix for the operator the time-dependent functions couple to, as given in the `dynamic` list used to define the Hamiltonian, here denoted by `Hd`. In the very end, we multiply the final output vector by the Schrödinger $-i$, which ensures the unitarity of real-time evolution.

```
##### real-time evolution of GPE #####
def GPE(time,psi):
    """
    This function solves the complex-valued time-dependent GPE:
    $$ i\dot\psi(t) = Hsp(t)\psi(t) + U |\psi(t)|^2 \psi(t) $$
    """
    # solve static part of GPE
    psi_dot = Hsp.static.dot(psi) + U*np.abs(psi)**2*psi
    # solve dynamic part of GPE
    for f,Hd in iteritems(Hsp.dynamic):
        psi_dot += f(time)*Hd.dot(psi)
    return -1j*psi_dot
# define real time vector
t=np.linspace(0.0,t_ramp,101)
```

To perform the real-time evolution explicitly we once again use the `evolve()` function. This time, however, since the solution of the GPE is anticipated to be complex-valued, and because we do not do imaginary time, we do not need to pass the flags `real` and `imag_time`. Instead, we decided to show the flags for the relative and absolute tolerance of the solver.

```
# time-evolve state according to GPE
psi_t = evolve(psi0,t[0],t,GPE,iterate=True,atol=1E-12,rtol=1E-12)
```

Finally, we can enjoy the “movie" displaying real-time evolution

```
# display state evolution
for i,psi in enumerate(psi_t):
    # compute energy
    E=(Hsp.matrix_ele(psi,psi,time=t[i]) + 0.5*U*np.sum(np.abs(psi)**4) ).real
    # compute trap
    kappa_trap=ramp(t[i],kappa_trap_i,kappa_trap_f,t_ramp)*(sites)**2
    # plot wave function
    plt.plot(sites, abs(psi0)**2, color='r',marker='s',alpha=0.2
                                ,label='$|\\psi_{\\mathrm{GS},j}|^2$')
    plt.plot(sites, abs(psi)**2, color='b',marker='o',label='$|\\psi_j(t)|^2$')
```

```
    plt.plot(sites, kappa_trap,'--',color='g',label='$\\mathrm{trap}$')
    plt.ylim([-0.01,max(abs(psi0)**2)+0.01])
    plt.xlabel('$\\mathrm{lattice\\ sites}$',fontsize=18)
    plt.title('$Jt=%0.2f,\\ E(t)-E_\\mathrm{GS}=%0.4fJ$'%(t[i],E-E_GS),fontsize
=18)
    plt.legend(loc='upper right',fontsize=18)
    plt.draw() # draw frame
    plt.pause(0.00005) # pause frame
    plt.clf() # clear figure
plt.close()
```

The complete code including the lines that produce Fig. 5 is available under:

http://weinbe58.github.io/QuSpin/examples/example8.html

### 2.6 Integrability Breaking and Thermalising Dynamics in the Translationally Invariant 2D Transverse-Field Ising Model

This example shows how to:

- define symmetries for 2D systems,
- use the `spin_basis_general` class to define custom symmetries,
- use the `obs_vs_time` routine with custom user-defined generator to calculate the expectation value of operators as a function of time,
- use the new functionality of the basis class to calculate the entanglement entropy.

*Physics Setup*— In Sec. 2.1, we introduced the one-dimensional transverse-field Ising (TFI) model and showed how one can map it to an exactly-solvable quadratic fermion Hamiltonian using the Jordan-Wigner transformation. This transformation allows one to obtain an exact closed-form analytic expression for the eigenstates and eigenenergies of the Hamiltonian (at least when the system obeys translational invariance). The fact that such a transformation exists, is intrinsically related to the model being integrable. Integrable models feature an extensive amount of local integrals of motion – conserved quantities which impose specific selection rules for the transitions between the many-body states of the system. As a result, such models thermalise in a restricted sense, i.e. in conformity with all these integrals of motion, and the long-time behaviour of the system is captured by a Generalised Gibbs Ensemble [31]. The system thermalises but thermalisation is constrained. On the other hand, non-integrable systems have a far less restricted phase space, and feature unconstrained thermalisation. Their long-time behaviour is, therefore, captured by the Gibbs Ensemble. Since thermalising dynamics appears to be intrinsically related to the lack of integrability, if detected, one can use it as an indicator for the absence of a simple, closed form expression for the eigenstates and the eigenenergies of generic quantum many-body systems.

Imagine you are given a model, and you want to determine its long-time thermalisation behaviour, from which in general you can infer information about integrability. One way to proceed is to conceive a numerical experiment as follows: if we subject the system to periodic driving at an intermediate driving frequency, this will break energy conservation. Then, in the non-integrable case, the system will keep absorbing energy from the drive until the state reaches infinite-temperature. On the contrary, in the integrable scenario unlimited energy absorption will be inhibited by the existence of additional conservation laws, and the system is likely to get stuck at some energy density. Hence, the long-time dynamics of out-of-equilibrium

many-body systems can be used to extract useful information about the complexity of the underlying Hamiltonian.

Below, we consider two models with periodic boundary conditions: (i) the 1d transverse field Ising model , and (ii) the 2d transverse field Ising model on a square lattice. There is no known simple mapping to a quadratic Hamiltonian in $2d$ and, therefore, the 2d Ising model is generally considered to no longer be integrable. Here, we study thermalization through energy absorption by periodically driving the two spin systems, and demonstrate the difference in the long-time heating behavior between the two. To this end, let us define two operators

$$H_{zz} = -\sum_{\langle ij\rangle} S_i^z S_j^z, \qquad H_x = -\sum_i S_i^x, \tag{16}$$

where $\langle ij\rangle$ restricts the sum over nearest neighbours, and use them to construct the following piecewise periodic Hamiltonian $H(t) = H(t+T)$:

$$H(t) = \begin{cases} H_{zz} + AH_x, & t \in [0, T/4), \\ H_{zz} - AH_x, & t \in [T/4, 3T/4), \\ H_{zz} + AH_x, & t \in [3T/4, T), \end{cases} \tag{17}$$

with $T$ the period of the drive, and $A$ and $\Omega = 2\pi/T$ – the corresponding driving amplitude and frequency, respectively. The dynamics is initialised in the ferromagnetic ground state of $H_{zz}$, $|\psi_i\rangle$. As the Hamiltonian obeys translation, parity and spin-inversion symmetry at all times, we will use this to reach larger system sizes by working in the symmetry sector which contains the ground state. We are only interested in the evolution of the state at integer multiples of the driving period $T$, i.e. stroboscopically. The state at the $\ell$-th period is conveniently obtained by successive application of the Floquet unitary $U_F$:

$$|\psi(\ell T)\rangle = [U_F(T)]^\ell |\psi_i\rangle, \qquad U_F(T) = \mathrm{e}^{-iT/4(H_{zz}+AH_x)}\mathrm{e}^{-iT/2(H_{zz}-AH_x)}\mathrm{e}^{-iT/4(H_{zz}+AH_x)}. \tag{18}$$

In order to see the difference in the long-time heating behaviour between $1d$ and $2d$, we measure the expectation value of $H_{zz}$ as a function of time. Let us define the relative energy $Q$ absorbed from the drive:

$$Q(t) = \left\langle \psi(t) \left| \frac{(H_{zz} - E_{\text{min}})}{E_{\text{inf. temp.}} - E_{\text{min}}} \right| \psi(t) \right\rangle = \left\langle \psi(t) \left| \frac{H_{zz} - E_{\text{min}}}{-E_{\text{min}}} \right| \psi(t) \right\rangle, \tag{19}$$

where the last equality comes from the fact that every state in the spectrum of $H_{zz}$ has a partner of opposite energy, and thus $E_{\text{inf. temp.}} = 0$. This quantity is normalised such that an infinite-temperature state has $Q = 1$, while a zero-temperature state: $Q = 0$.

Since $H_{zz}$ is not the only available observable, one might worry that some of the information contained in the quantum state is not captured in $Q(t)$. Hence, we also look at the entanglement entropy density (which is a property of the quantum state itself)

$$s_{\text{ent}}(t) = -\frac{1}{|\mathrm{A}|}\mathrm{tr}_{\mathrm{A}}\left[\rho_{\mathrm{A}}(t)\log\rho_{\mathrm{A}}(t)\right], \qquad \rho_{\mathrm{A}}(t) = \mathrm{tr}_{\mathrm{A^c}}|\psi(t)\rangle\langle\psi(t)| \tag{20}$$

of subsystem A, defined to contain the left half of the chain, and the bottom half of the system in the case of the 2D version. We denoted the reduced density matrix of subsystem A by $\rho_{\mathrm{A}}$, and $\mathrm{A^c}$ is the complement of A. To obtain a dimensionless quantity, we normalise the entanglement entropy by the corresponding Page value [99].

The dynamics of the excess energy and the entanglement are shown in Fig. 6.

*Code Analysis*—The code snippet below explains how to carry out the proposed study using QuSpin. We assume the reader has basic familiarity with QuSpin to set up simple time-dependent Hamiltonians. As is customary, we begin by loading the required Python libraries and packages:

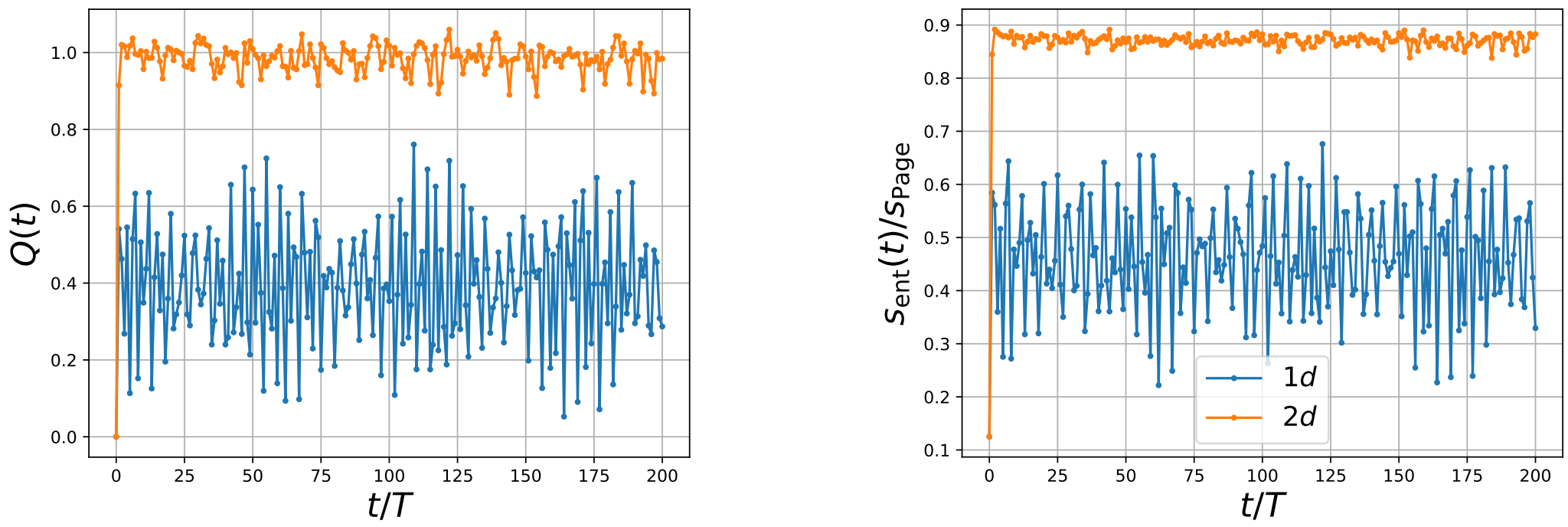

(a) $Q(t)$ for $t = \ell T$

(b) $s_{\rm ent}(t)/s_{\rm Page}$ for $t = \ell T$

Figure 6: Comparing the dynamics of $Q(t)$ (a) and $S_{\rm ent}(t)$ (b) for $1d$ (orange) and $2d$ (blue) at stroboscopic times ($t = nT$). Both systems contain $N = 16$ sites to make sure the many-body Hilbert spaces have roughly the same number of states. $s_{\rm ent}$ is normalized by the Page entropy per site. The drive frequency is $\Omega = 4$.

```
from __future__ import print_function, division
from quspin.operators import hamiltonian, exp_op # operators
from quspin.basis import spin_basis_1d, spin_basis_general # spin basis
    constructor
from quspin.tools.measurements import obs_vs_time # calculating dynamics
from quspin.tools.Floquet import Floquet_t_vec # period-spaced time vector
import numpy as np # general math functions
import matplotlib.pyplot as plt # plotting library
```

Then, we define the model parameters. Note that we choose all physical couplings to be unity, so the only parameters are the drive frequency and the drive amplitude, which is set equal to the driving frequency [the value of which is chosen arbitrarily]. The variable L denotes the linear dimension of the two spin-1/2 systems: the $1d$ chain has a total of L_1d sites, while the $2d$ square lattice has linear dimensions Lx and Ly, and a total number of sites N_2d. All variables indexed by _1d and _2d in the code below refer to the $1d$ and $2d$ system, respectively.

```
###### define model parameters ######
L_1d = 16 # length of chain for spin 1/2
Lx, Ly = 4, 4 # linear dimension of spin 1 2d lattice
N_2d = Lx*Ly # number of sites for spin 1
Omega = 2.0 # drive frequency
A = 2.0 # drive amplitude
```

Before we move on to define the two bases, let us explain how to use the spin_basis_general class two manually define custom symmetries in $2d$. QuSpin can handle any unitary symmetry operation $Q$ of multiplicity $m_Q$ ($Q^{m_Q} = 1$). We label its eigenvalues $\exp(-2\pi i q/m_Q)$ by an integer $q = \{0, 1, \dots, m_Q - 1\}$, which is used to set the symmetry block. For instance, if $Q = P$ is parity (reflection), then $q = 0, 1$ will correspond to the positive and negative parity blocks. If $Q = T$ is translation, then $q$ will label all momentum blocks. In the present case, the symmetries are translations and parity along the $x$ and $y$ lattice directions, and spin inversion. To this end, we first label all sites with integers, and store them in the variable s. One can think of the $2d$ system as a snake-like folded (or square-reshaped) s. With this representation each site s can be mapped to a ix and iy coordinate by the mapping: s=ix+Lx*iy. The action of any local symmetry can be programmed on this set of sites. For instance, this helps us define the action of the $x$ and $y$ translation operators T_x and T_y, by shifting all sites in the corresponding direction. Parity (reflection) operations P_x and P_y are also intuitive to define. Last, the

spin inversion symmetry is uniquely defined through the mapping `s` ↦ `-(1+s)` for every site `s` it acts on. In this way, the user can define different lattice geometries, and/or sublattice symmetries and QuSpin will do the hard job to reduce the Hilbert space to the corresponding symmetry sector.

```python
###### setting up user-defined symmetry transformations for 2d lattice ######
s = np.arange(N_2d) # sites [0,1,2,....]
x = s%Lx # x positions for sites
y = s//Lx # y positions for sites
T_x = (x+1)%Lx + Lx*y # translation along x-direction
T_y = x +Lx*((y+1)%Ly) # translation along y-direction
P_x = x + Lx*(Ly-y-1) # reflection about x-axis
P_y = (Lx-x-1) + Lx*y # reflection about y-axis
Z   = -(s+1) # spin inversion
```

Let us define the spin bases. For the $1d$ case, we make use of the `spin_basis_1d` constructor. We note in passing that one can also study higher-spin systems by using the optional argument `S`, which accepts a string (integer or half-integer) to specify the spin vector size, see documentation C. Requesting symmetry blocks works as usual, by using the corresponding optional arguments. For the $2d$ spin system, we use the `spin_basis_general` class. This constructor allows the user to build basis objects with many user-defined symmetries, based on their action on the lattice sites. Unlike the $1d$-basis constructors, that have symmetry blocks with pre-defined variables which take integer values `kblock=0`, the general basis constructors accept user-defined block variable names which take a tuple for every symmetry requested: the first entry is the symmetry transformation itself, and the second one – the integer which labels the required symmetry block, e.g. `kxblock=(T_x,0)`.

```python
###### setting up bases ######
basis_1d = spin_basis_1d(L_1d,kblock=0,pblock=1,zblock=1) # 1d - basis
basis_2d = spin_basis_general(N_2d,kxblock=(T_x,0),kyblock=(T_y,0),
                pxblock=(P_x,0),pyblock=(P_y,0),zblock=(Z,0)) # 2d - basis
# print information about the basis
print("Size of 1D H-space: {Ns:d}".format(Ns=basis_1d.Ns))
print("Size of 2D H-space: {Ns:d}".format(Ns=basis_2d.Ns))
```

To set up the site-coupling lists for the two operators in the Hamiltonian we proceed in the usual manner. Using them, we can call the `hamiltonian` constructor to define the operators $-\sum_{\langle ij\rangle} S^z_i S^z_j$ and $-\sum_j S^x_j$. Here we keep the operators separate, in order to do the periodic step-drive evolution, which is why we do not need to define separate static and dynamic lists.

```python
###### setting up operators in hamiltonian ######
# setting up site-coupling lists
Jzz_1d=[[-1.0,i,(i+1)%L_1d] for i in range(L_1d)]
hx_1d =[[-1.0,i] for i in range(L_1d)]
#
Jzz_2d=[[-1.0,i,T_x[i]] for i in range(N_2d)]+[[-1.0,i,T_y[i]] for i in range(
    N_2d)]
hx_2d =[[-1.0,i] for i in range(N_2d)]
# setting up hamiltonians
# 1d
Hzz_1d=hamiltonian([["zz",Jzz_1d]],[],basis=basis_1d,dtype=np.float64)
Hx_1d =hamiltonian([["x",hx_1d]],[],basis=basis_1d,dtype=np.float64)
# 2d
Hzz_2d=hamiltonian([["zz",Jzz_2d]],[],basis=basis_2d,dtype=np.float64)
Hx_2d =hamiltonian([["x",hx_2d]],[],basis=basis_2d,dtype=np.float64)
```

In order to do time evolution, we need to define the initial state of our system. In this case, we start from the ground state of the Hamiltonian $-\sum_{\langle ij\rangle} S^z_i S^z_j$. We remind the reader that, since

we work in a specific symmetry sector, this state may no longer be a product state. To this end, we employ the `eigsh()` method for sparse hermitian matrices of the `hamiltonian` class, where we explicitly specify that we are interested in getting a single (`k=1`) smallest algebraic (i.e. ground state) eigenenergy, and the corresponding eigenstate (a.k.a. the ground state), c.f. Example 0 in Ref. [16] for more details.

```
###### calculate initial states ######
# calculating bandwidth for non-driven hamiltonian
[E_1d_min],psi_1d = Hzz_1d.eigsh(k=1,which="SA")
[E_2d_min],psi_2d = Hzz_2d.eigsh(k=1,which="SA")
# setting up initial states
psi0_1d = psi_1d.ravel()
psi0_2d = psi_2d.ravel()
```

We are now set to do study the dynamics following the periodic step-drive. Before we go into the details, we note that QuSpin contains a build-in `Floquet` class under the `tools` module which can be useful for studying this and other periodically-driven systems, see Example 2 from Ref. [16]. Here, instead, we focus on manually evolving the state. First, we define the number of periods we would like to stroboscopically evolve our system for. Stroboscopic evolution is one where all quantities are evaluated at integer multiple of the driving period. To set up a time vector, which explicitly hits all those points, we use the `Floquet_t_vec` class, which accepts as arguments the frequency `Omega`, the number of periods `nT`, and the number of points per period `len_T`. The `Floquet_t_vec` class creates an object which has many useful attributes, including the stroboscopic times and their indices, the period, the starting point, etc. We invite the interested reader to check out the documentation for more information C.

```
###### time evolution ######
# stroboscopic time vector
nT = 200 # number of periods to evolve to
t=Floquet_t_vec(Omega,nT,len_T=1) # t.vals=t, t.i=initial time, t.T=drive period
```

Since the Hamiltonian is piece-wise constant, we can simulate the time evolution by exponentiating the separate terms. Note that, since we choose the driving phase (Floquet gauge) to yield a time-symmetric Hamiltonian, i.e. $H(-t) = H(t)$, this results in evolving the system with the Hamiltonians $H_{zz} + AH_x$, $H_{zz} - AH_x$, $H_{zz} + AH_x$ for the durations $T/4, T/2, T/4$, respectively (think of the phase of the drive as that of a rectilinear cosine drive $\operatorname{sgn}\cos\Omega t$). To compute the matrix exponential of a static operator, we make use of the `exp_op` class, where $\exp(zB)$ =`exp_op(B,a=z)` for some complex number $z$ and some operator $B$.

```
# creating generators of time evolution using exp_op class
U1_1d = exp_op(Hzz_1d+A*Hx_1d,a=-1j*t.T/4)
U2_1d = exp_op(Hzz_1d-A*Hx_1d,a=-1j*t.T/2)
U1_2d = exp_op(Hzz_2d+A*Hx_2d,a=-1j*t.T/4)
U2_2d = exp_op(Hzz_2d-A*Hx_2d,a=-1j*t.T/2)
```

In order to evolve the state itself, we demonstrate how to construct a user-defined generator function `evolve_gen()`, which takes the initial state, the number of periods, and a sequence of unitaries within a period to apply them on the state. The generator character of the function means that it will not execute the loops it contains when called for the first time, but rather store information about them, and return the values one by one when prompted to do so later on. This is useful since otherwise we would have to loop over all times to evolve the state first, and then once again to compute the observables. As we can we below, the generator function allows us to get away with a single loop.

```
# user-defined generator for stroboscopic dynamics
def evolve_gen(psi0,nT,*U_list):
    yield psi0
```

```
    for i in range(nT): # loop over number of periods
        for U in U_list: # loop over unitaries
            psi0 = U.dot(psi0)
        yield psi0
# get generator objects for time-evolved states
psi_1d_t = evolve_gen(psi0_1d,nT,U1_1d,U2_1d,U1_1d)
psi_2d_t = evolve_gen(psi0_2d,nT,U1_2d,U2_2d,U1_2d)
```

Finally, we are ready to compute the time-dependent quantities of interest. In order to calculate the expectation $\langle\psi(t)|H_{zz}|\psi(t)\rangle$ QuSpin has a routine called `obs_vs_time()`. It accepts the time-dependent state `psi_12_t` (or its generator), the time vector `t.vals` to evaluate the observable at, and a dictionary, which contains all observables of interest (here `Hzz_12`). The output of `obs_vs_time()` is a dictionary which contains the results: every observable is being parsed by a unique key (string) (here `"E"`), under which its expectation value will appear, evaluated at the requested times. Further, if one specifies the optional argument `return_state=True`, the time-evolved state is also returned under the key `"psi_t"`.

```
###### compute expectation values of observables ######
# measure Hzz as a function of time
Obs_1d_t = obs_vs_time(psi_1d_t,t.vals,dict(E=Hzz_1d),return_state=True)
Obs_2d_t = obs_vs_time(psi_2d_t,t.vals,dict(E=Hzz_2d),return_state=True)
```

In fact, `obs_vs_time()` can also compute the entanglement entropy at every point of time (see documentation or Sec. 2.7). Instead, we decided to show how one can do this using the new functionality of the `basis` class. Each basis constructor comes with a function method `ent_entropy()` which evaluates the entanglement entropy of a given state, and may return the reduced density matrix upon request. To compute the entanglement, the user needs to pass the state (here `Obs_12_t["psi_t"]`), and a subsystem to define the partition for computing the entanglement. The method `ent_entropy()` can handle vectorised calculations, and will compute the entanglement of the state for each point of time. The output is stored in a dictionary, and the entanglement entropy can be accessed with the key `"Sent_A"`. Finally, to obtain the entanglement entropy density, we also normalise the results by the size of the subsystem of interest.

```
# calculating the entanglement entropy density
Sent_time_1d = basis_1d.ent_entropy(Obs_1d_t["psi_t"],sub_sys_A=range(L_1d//2))["Sent_A"]/(L_1d//2)
Sent_time_2d = basis_2d.ent_entropy(Obs_2d_t["psi_t"],sub_sys_A=range(N_2d//2))["Sent_A"]/(N_2d//2)
```

In order get an intuition about the amount of entanglement generated in the system by the drive, we use as a reference entanglement the corresponding Page values, which is the average amount of entanglement between two subsystems of a random vectors in the many-body Hilbert space.

```
# calculate entanglement entropy density
s_p_1d = np.log(2)-2.0**(-L_1d//2-L_1d)/(2*(L_1d//2))
s_p_2d = np.log(2)-2.0**(-N_2d//2-N_2d)/(2*(N_2d//2))
```

The complete code including the lines that produce Fig. 6 is available in under:

http://weinbe58.github.io/QuSpin/examples/example9.html

### 2.7 Out-of-Equilibrium Bose-Fermi Mixtures

The last example in our tutorial shows how to

- construct Hamiltonians for Bose-Fermi mixtures using the `tensor_basis` class,
- periodically drive one subsystem (here the fermions),
- use new `tensor_basis.index()` functionality to construct simple product states in the tensor basis,
- use `obs_vs_time` functionality to compute the evolution of the entanglement entropy of the bosons with the fermions.

*Physics Setup*—The interest in the Bose-Fermi Hubbard (BFH) model is motivated from different areas of condensed matter and atomic physics. Studying the dressing of (interacting) fermionic atoms submerged in a superfluid Bose gas [100], the study of the sympathetic cooling technique [101] to cool down spin-polarised fermions which do not interact in the $s$-wave channel, etc., are only a few of the experimental platforms for the rich physics concealed by Bose-Fermi mixtures (BFM). On the theoretical side, the BFH model is seen as a playground for the understanding of exotic phases of matter [102–105], such as the coexistence of superfluid and checkerboard order, supersolid states, and the emergence of dressed compound particles. It is also a natural candidate for the search of manifestations of supersymmetry in condensed matter.

In this section, we study the generation of interspecies entanglement in a spinless Bose-Fermi mixture, caused by an external time-dependent drive. The Hamiltonian for the system reads

$$
\begin{aligned}
H(t) &= H_\mathrm{b} + H_\mathrm{f}(t) + H_\mathrm{bf},\\
H_\mathrm{b} &= -J_\mathrm{b}\sum_j\left(b^\dagger_{j+1}b_j + \mathrm{h.c.}\right) - \frac{U_\mathrm{bb}}{2}\sum_j n^\mathrm{b}_j + \frac{U_\mathrm{bb}}{2}\sum_j n^\mathrm{b}_j n^\mathrm{b}_j,\\
H_\mathrm{f}(t) &= -J_\mathrm{f}\sum_j\left(c^\dagger_{j+1}c_j - c_{j+1}c^\dagger_j\right) + A\cos\Omega t\sum_j(-1)^j n^\mathrm{f}_j + U_\mathrm{ff}\sum_j n^\mathrm{f}_j n^\mathrm{f}_{j+1},\\
H_\mathrm{bf} &= U_\mathrm{bf}\sum_j n^\mathrm{b}_j n^\mathrm{f}_j,
\end{aligned}
\tag{21}
$$

where the operator $b^\dagger_j$ ($c^\dagger_j$) creates a boson (fermion) on site $j$, and the corresponding density is $n^\mathrm{b}_j = b^\dagger_j b_j$ ($n^\mathrm{f}_j = c^\dagger_j c_j$). The hopping matrix elements are denoted by $J_\mathrm{b}$ and $J_\mathrm{f}$, respectively. The bosons are subject to an on-site interaction of strength $U_\mathrm{bb}$, while the spin-polarised fermion-fermion interaction $U_\mathrm{ff}$ is effective on nearest-neighbouring sites. The bosonic and fermionic sectors are coupled through an on-site interspecies density-density interaction $U_\mathrm{bf}$. We assume unit filling for the bosons and half-filling for the fermions.

The BF mixture is initially prepared in the product state $|b\rangle|f\rangle = \prod_{j=0}^{L-1} b^\dagger_j|0\rangle \prod_{j=0}^{L/2-1} c^\dagger_j|0\rangle$, which is a Mott state for the bosons, and a domain wall for the fermions. A low-frequency periodic drive of amplitude $A$ and frequency $\Omega$ couples to the staggered potential in the fermions sector, and pumps energy into the system. We study the growth of the entanglement $S_\mathrm{ent}(t)$ between the two species, see Fig. 7.

$$
S_\mathrm{ent}(t) = -\mathrm{tr}_\mathrm{b}\left(\rho_\mathrm{b}(t)\log\rho_\mathrm{b}(t)\right), \quad \rho_\mathrm{b}(t) = \mathrm{tr}_\mathrm{f}|\psi(t)\rangle\langle\psi(t)|,
\tag{22}
$$

where $\mathrm{tr}_\mathrm{b}(\cdot)$ and $\mathrm{tr}_\mathrm{f}(\cdot)$ are the traces over the boson and fermion sectors, respectively.
*Code Analysis*—Let us explain how to code up the Hamiltonian for the BHM now. As always, we begin by loading the necessary modules for the simulation. New here is the `tensor_basis` class with the help of which one can construct the basis for a tensor product Hilbert space:

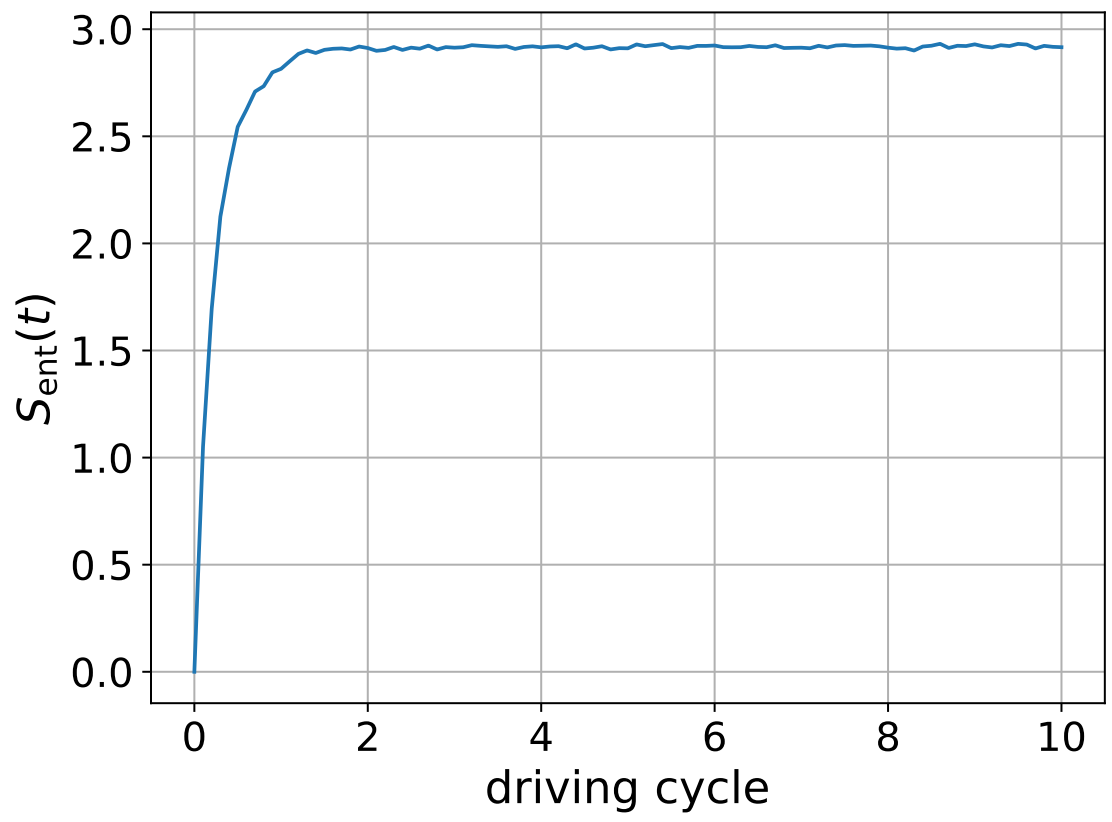

Figure 7: Entanglement entropy of one species as a function of time. This data was taken on a chain of length $L = 6$, which agrees with the Page value [99] $S_{\text{page}} \approx 2.86$.

```
from __future__ import print_function, division
from quspin.operators import hamiltonian # Hamiltonians and operators
from quspin.basis import tensor_basis,spinless_fermion_basis_1d,boson_basis_1d # bases
from quspin.tools.measurements import obs_vs_time # calculating dynamics
from quspin.tools.Floquet import Floquet_t_vec # period-spaced time vector
import numpy as np # general math functions
import matplotlib.pyplot as plt # plotting library
```

First, we define the model parameters, and the drive:

```
##### setting up parameters for simulation
# physical parameters
L = 6 # system size
Nf, Nb = L//2, L # number of fermions, bosons
N = Nf + Nb # total number of particles
Jb, Jf = 1.0, 1.0 # boson, fermon hopping strength
Uff, Ubb, Ubf = -2.0, 0.5, 5.0  # bb, ff, bf interaction
# define time-dependent perturbation
A = 2.0
Omega = 1.0
def drive(t,Omega):
    return np.sin(Omega*t)
drive_args=[Omega]
```

Next we set up the basis, introducing the `tensor_basis` constructor class. In its full-fledged generality, `tensor_basis` takes $n$ basis objects which it uses to construct the matrix elements in the tensor product space:

$$\mathcal{H} = \mathcal{H}_1 \otimes \mathcal{H}_2 \otimes \cdots \otimes \mathcal{H}_n. \tag{23}$$

Here we consider the case of two Hilbert spaces, $\mathcal{H} = \mathcal{H}_1 \otimes \mathcal{H}_2$, one for the bosons and one for fermions[7]. One disadvantage of the tensor basis is that it does not allow for the use of symmetries, beyond particle-conservation (magnetisation in the case of spin systems). This is because a tensor basis need not obey the symmetries of the individual bases.

```
###### create the basis
# build the two bases to tensor together to a bose-fermi mixture
basis_b=boson_basis_1d(L,Nb=Nb,sps=3) # boson basis
basis_f=spinless_fermion_basis_1d(L,Nf=Nf) # fermion basis
```

[7] to construct a `tensor_basis` object in general: `t_basis = tensor_basis(basis_1,basis_2,...,basis_n)`

```
basis=tensor_basis(basis_b,basis_f) # BFM
```

To create the Hamiltonian, we again use the usual form of the site-coupling lists, as if we would construct a single-species operator. Since the site-coupling lists do not yet know which operators they will refer to, this is straightforward [mind the signs for the fermionic hopping operators, though, see Sec. 2.2]. We use the subscripts `b`, `f` and `bf` to designate which species this lists will refer to:

```
##### create model
# define site-coupling lists
hop_b = [[-Jb,i,(i+1)%L] for i in range(L)] # b hopping
int_list_bb = [[Ubb/2.0,i,i] for i in range(L)] # bb onsite interaction
int_list_bb_lin = [[-Ubb/2.0,i] for i in range(L)] # bb interaction, linear term
#
hop_f_right = [[-Jf,i,(i+1)%L] for i in range(L)] # f hopping right
hop_f_left = [[Jf,i,(i+1)%L] for i in range(L)] # f hopping left
int_list_ff = [[Uff,i,(i+1)%L] for i in range(L)] # ff nearest-neighbour interaction
drive_f = [[A*(-1.0)**i,i] for i in range(L)] # density staggered drive
#
int_list_bf = [[Ubf,i,i] for i in range(L)] # bf onsite interaction
```

The new part comes in specifying the `static` and `dynamic` lists. This is where we tell QuSpin that we are dealing with two species tensored together. Notice here that the `"|"` character is used to separate the operators which belong to the boson (left side of tensor product) and fermion (right side of tensor product) Hilbert spaces in `basis`. If no operator string in present, the operator is assumed to be the identity `'I'`. The site-coupling lists, on the other hand, do not require separating the two sides of the tensor product, as it is assumed that the operator string lines up with the correct site index when the `'|'` character is removed. For instance, the only term which couples the bosons and the fermions is a density-density interaction. It is the corresponding site-coupling list, thought, that determines that it is of on-site type. This type of syntax is the same for both `static` and `dynamic` lists.

```
# create static lists
static = [
            ["+-|", hop_b], # bosons hop left
            ["-+|", hop_b], # bosons hop right
            ["n|", int_list_bb_lin], # bb onsite interaction
            ["nn|", int_list_bb], # bb onsite interaction
            #
            ["|+-", hop_f_left], # fermions hop left
            ["|-+", hop_f_right], # fermions hop right
            ["|nn", int_list_ff], # ff nn interaction
            #
            ["n|n", int_list_bf], # bf onsite interaction
            ]
dynamic = [["|n",drive_f,drive,drive_args]] # drive couples to fermions only
```

Computing the Hamiltonian for the BHM is done in one line:

```
###### set up Hamiltonian and initial states
no_checks = dict(check_pcon=False,check_symm=False,check_herm=False)
H_BFM = hamiltonian(static,dynamic,basis=basis,**no_checks)
```

Since the basis states are coded as integers, it might be hard to find the integer corresponding to a particular Fock basis state. Therefore, the `tensor_basis` class has a method for finding the index of a particular Fock state in the `basis`. The user just has to pass a string with zeros and ones to determine which sites are occupied and which empty. For the BHM, we choose an

initial state where each site is occupied by one boson, while the fermions start in a domain wall occupying the left half of the chain. We call the index of the basis array which corresponds to the initial state `i_0`. To create the pure initial Fock state in the full Hilbert space, we first define an array `psi_0` which is empty, except for the position `i_0`, where the component is set to unity by the wavefunction normalisation requirement.

```
# basis.index accepts strings and returns the index which corresponds to that
    state in the basis list
i_0 = basis.index(s_b,s_f) # find index of product state in basis
psi_0 = np.zeros(basis.Ns) # allocate space for state
psi_0[i_0] = 1.0 # set MB state to be the given product state
print("H-space size: {:d}, initial state: |{:s}>|{:s}>".format(basis.Ns,s_b,s_f))
```

To compute the time evolution of the state under the Hamiltonian $H(t)$, we make use of the `Floquet_t_vec` class to create a time vector which hits all stroboscopic points, and further contains `len_T=10` points per period. For the dynamics, we use the `evolve()` method of the `hamiltonian` constructor class (the details of this have been explain in Sec. 2.2). Note that we create a generator object `psi_t`, which yields the evolved state, time step by time step.

```
###### time evolve initial state and measure entanglement between species
t=Floquet_t_vec(Omega,10,len_T=10) # t.vals=times, t.i=initial time, t.T=drive
    period
psi_t = H_BFM.evolve(psi_0,t.i,t.vals,iterate=True)
```

In Sec. 2.6, we showed how one can use the method `basis.ent_entropy()` to compute the time evolution of the entanglement entropy, if we have the time-evolved state. Here, we show a different way of for doing the same. We use the `measurements` function `obs_vs_time()`, with the user-defined generator `psi_t` as an input. In general, `obs_vs_time()`, calculates the expectation of observables, see Sec. 2.6. However, if we parse a non-empty dictionary, called `Sent_args`, which contains the arguments for `basis.ent_entropy()`, we can immediately get the result directly out of the generator.

```
# measure observable
Sent_args=dict(basis=basis,sub_sys_A="left")
meas = obs_vs_time(psi_t,t.vals,{},Sent_args=Sent_args)
```

The output of `obs_vs_time()` is a dictionary. The results pertaining to the calculation of entanglement are stored under the key `"Sent_time"`. The corresponding value itself is another dictionary, with the output of `basis.ent_entropy()`, in which the entropy is stored under the key `"Sent_A"`. That second dictionary can also contain other objects, such as the reduced density matrix evaluated at all time steps, if these are specified in the variable `Srdm_args`. For more information on that, we refer the user to the documentation.

```
# read off measurements
Entropy_t = meas["Sent_time"]["Sent_A"]
```

The complete code including the lines that produce Fig. 7 is available under:

http://weinbe58.github.io/QuSpin/examples/example10.html

# 3 New Horizons for QuSpin

As we demonstrated using various examples, QuSpin is currently capable of simulating a huge class of dynamical quantum systems. Nonetheless, we do not consider it a complete project, as we imagine adding further different functionalities, motivated by the needs of the users.

Interested users can put the project on their Github watch list, which will notify them of any new releases.

Being an open project, we invite the community to fork the Github QuSpin repository, and actively contribute to the development of the project! We would be more than happy to consider well-documented functions and classes, which build on QuSpin, and include them in further official releases so they can be used by the wider community. We will also consider patches which allow to combine QuSpin with other open-source packages for studying quantum physics. If you have ideas, just email us, or contact us on Github.

We would also much appreciate it, if users can properly report every single instance of a bug or malfunction in the issues section of the project repository, as this is an inevitable part of maturing the code.

***Parallel Capabilities.—*** Starting from version 0.3.2, QuSpin has full support for parallel computation:

- OpenMP support allows to speed up calculations by parallelizing low-level loops using multiple threads. Many QuSpin functions take advantage of this within the `hamiltonian` and `basis` constructor classes, including (but not limited to) `hamiltonian.evolve()`, `hamiltonian.dot()`, `basis.Op()`, `tools.evolve.expm_multiply_parallel()`, etc, as well as constructing `basis` and `hamiltonian` objects themselves. It is important to mention that, in order to make use of OpenMP support, the user needs to install the `omp` version of QuSpin, which is different from the standard version. Please check http://weinbe58.github.io/QuSpin/parallelization.html for more information on how to install and use OpenMP in QuSpin.

- support of Intel's MKL linrary is inherited automatically, since QuSpin wraps some of the Numpy functionality. While this feature is not intrinsic to QuSpin per se (though it is available in QuSpin via Numpy), it can help speeding up diagonalization functions, such as `hamiltonian.eigh()` and `hamiltonian.eigsh()`, etc. Note that currently the default version of Numpy installed with anaconda has MKL support built-in.

If you are interested in speeding up your QuSpin computation and you have multiple cores available at your disposal (e.g., on a modern desktop/laptop computer, or better yet on a computing cluster), check out Example 12 online. The developers of QuSpin realize that GPU computing is becoming more important as the GPU hardware is improving rapidly compared to CPU hardware. There are a few libraries which mimic the functionality of NumPy and SciPy on GPUs. Much of that being the linear algebra operations for both sparse and dense matrices [8]. Many of the features in these libraries can be access by converting QuSpin objects to NumPy/SciPy objects which can then interact with the GPU library. However, there are two important reasons we are not directly including these libraries into QuSpin: the first reason being a lack of functionality for numerical integration of ordinary differential equations which is necessary for solving the time-dependent Schrödinger equation, and the second reason being that some of these libraries are are not hosted on Anaconda, which makes it impossible to use them in the packaging process for QuSpin.

Additional features we would like to consider in future versions of QuSpin include: a basis class dedicated to single particle lattice physics allowing for more efficient and intuitive way of implementing single particle Hamiltonians, functionality for doing Lindblad and other kinds of dynamics with density matrices, finite-temperature Lanzcos methods, class for calculating correlation functions, and overall improving the efficiency of current functionality.

[8] check out the package CuPy for a good example of this.

## Acknowledgements

We would like to thank S. Capponi, M. Kolodrubetz, S. Kourtis and L. Pollet for various stimulating discussions. Special thanks go to Alex G.R. Day for urging us to put a proper documentation for the package, and to W.W. Ho for reporting some bugs in the code. We also thank all users who reported various problems on Github, and encourage them to keep doing so in the future. The authors are pleased to acknowledge that the computational work reported on in this paper was performed on the Shared Computing Cluster which is administered by Boston University's Research Computing Services. The authors also acknowledge the Research Computing Services group for providing consulting support which has contributed to the results reported within this paper. We would also like to thank Github for providing the online resources to help develop and maintain this project, and the Sphinx project which makes it possible to maintain an easy-to-read and up-to-date documentation.

**Funding information** MB acknowledges support from the Emergent Phenomena in Quantum Systems initiative of the Gordon and Betty Moore Foundation, ERC synergy grant UQUAM, and the U.S. Department of Energy, Office of Science, Office of Advanced Scientific Computing Research, Quantum Algorithm Teams Program. This research was supported in part by the National Science Foundation under Grant No. NSF PHY-1748958.

## A Installation Guide in a Few Steps

Detailed installation instructions can be found under:

http://weinbe58.github.io/QuSpin/Installation.html

## B Basic Use of Command Line to Run Python

For beginners, instructions how to execute a Python code are provided under:

http://weinbe58.github.io/QuSpin/Installation.html

## C Package Documentation

In QuSpin quantum many-body operators are represented as matrices. The computation of these matrices are done through custom code written in Cython. Cython is an optimizing static compiler which takes code written in a syntax similar to Python, and compiles it into a highly efficient C/C++ shared library. These libraries are then easily interfaced with Python, but can run orders of magnitude faster than pure Python code [106]. The matrices are stored in a sparse matrix format using the sparse matrix library of SciPy [107]. This allows QuSpin to easily interface with mature Python packages, such as NumPy, SciPy, any many others. These packages provide reliable state-of-the-art tools for scientific computation as well as support from the Python community to regularly improve and update them [107–110]. Moreover, we have included specific functionality in QuSpin which uses NumPy and SciPy to do many desired calculations common to ED studies, while making sure the user only has to call a few

NumPy or SciPy functions directly. The complete up-to-date documentation for the package is available online under:

http://weinbe58.github.io/QuSpin/index.html

## D Complete Example Codes

The python scripts for all QuSpin examples can be downloaded under:

http://weinbe58.github.io/QuSpin/Examples.html

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
