# Peer review of "QuSpin: a Python Package for Dynamics and Exact Diagonalisation of Quantum Many Body Systems. Part II: bosons, fermions and higher spins"

_SciPost Physics, doi:SciPost Phys. 7, 020 (2019)_

## Round 1 · Referee Report · Anonymous (Referee 2) · 2018-6-12

Strengths

  1. Very useful package for a broad community
  2. Powerful new features: support for fermionic, bosonic, and higher spin systems, computation of entanglement entropies and spectra, the possibility to introduce user-defined symmetries, parameter-dependent Hamiltonians, and more
  3. Very useful tutorials with interesting practical examples
  4. Open-source! The authors encourage the community to get involved in the project.

Weaknesses

  1. The parallel capabilities of the package are not so clear

Report

In this paper the authors present an updated version of their open-source Python package called QuSpin for exact diagonalization and quantum dynamics for quantum lattice models. They explain their package in great detail based on different interesting examples in a tutorial-style way.

As already the previous version of the package, I find this project a great initiative which I believe is very useful for a broad community, ranging from undergraduate students to senior theory and experimental researchers. With this major update the authors have substantially increased the range of applicability of their package, making a new release absolutely justified.

For these reasons I can recommend publication of this article in its present form (up to minor details, see suggested changes).

Requested changes

Suggested changes: 1. The only point of improvement I see is to better comment on the parallel capabilities of the package which is not very clear from the text (there is only one remark on p22, and a few remarks on p 40 with reference to the Documentation C). It would be interesting to have a quick overview somewhere, to what extend the package can be used in parallel computations. (If certain parts do not support parallelism yet, that could be mentioned as possible feature in a future release). 2. a few typos and minor points: - p.8 title of Sec 2.2: write out SSH -> Su-Schrieffer-Heeger - Figure 4 caption rhs: (b) local -> (b) Local - p.24 after the first code block: adder -> ladder - Fig.5: figure labels are borderline small - p27, after code line 41: Hsmiltonian -> Hamiltonian - p41, first paragraph: Linblad -> Lindblad

---

## Round 1 · Referee Report · Seyed Nariman Saadatmand (Referee 1) · 2018-6-12

Strengths

1- QuSpin is well-documented and relatively easy to install; while, existing or new scripts for a desired physical model can be managed and executed conveniently. 2- Supporting arbitrary integer and half-integer spin, fermionic, bosonic, and mixed-species models. 3- Providing basis constructors that support a large class of physical symmetries, which allow users to run jobs with built-in Hamiltonian symmetries. In addition, for dynamical calculations, QuSpin can split the evolution over relevant symmetry sectors (i.e. project out a non-symmetric initial wavefunction into different symmetry sectors and, then, evolve each sector separately). 4- The ability to study free-particle translation-invariant Hamiltonians in real-space using single-particle states, in oppose to employing a Fock-space basis. This would allow users to significantly scale up system size due to the absence of the exponential scaling of the Hilbert space dimension for non-interacting models. 5- The ability to construct a (generator) list containing the operators in a series of requested times or the corresponding time-evolved states. 6- The ability to solve user-defined equations of motion (in practice, a series of ODEs). 7- Supporting tensor-product bases, which allows users to construct product states more easily. 8- Although QuSpin is nominally a Python package, its matrix computations are written in and done through Cython, which is a tool that compiles Python syntax to highly-efficient C/C++ routines, making the execution orders of magnitude faster than pure Python codes.

Weaknesses

1- In operator lists/strings, $S^x$ and $S^y$ operators are only available for spin-1/2 systems (corresponding to Pauli $\sigma^x$ and $\sigma^y$ operators). 2- Although the paper presents a large number of tutorials covering interesting physical models, it lacks an example discussing the construction of a geometrically frustrated spin Hamiltonian, such as Heisenberg-type models on honeycomb, Kagome, or triangular lattice. The mapping from the ''snake-like folded'' chain, parametrized by $s$, to the 2D lattice is slightly less trivial than the square-lattice model of the example in Sec. 2.6, and the emergent physics is rather more interesting and, also, more difficult for other numerical methods to tackle. 3- The following is only a potential weakness for the QuSpin core codes and not essentially relates the manuscript: during the execution of Python jobs, I wish there was more informative messages/warnings printed to stdout by default, especially while performing the actual diagonalization calculations (activating '-v' options of Python, will not help as it prints too many messages, mostly useless for getting information on the physically-important numerics).

Report

In this paper, the authors present the new features and several interesting tutorials for their open-source Python package, QuSpin [http://weinbe58.github.io/QuSpin/index.html]. After the recent major updates, QuSpin can be now considered as a comprehensive and well-optimized numerical toolkit to tackle the exact diagonalization (ED) and (imaginary) time-evolution calculations for quantum many-body systems. It is worth mentioning that performing such static and dynamical ED calculations are still relevant today (e.g. see the reasons discussed in the Requested Changes part of this report). The basic documentation and a series of rather less-complicated physical examples were presented in the package's first paper, [SciPost Phys. 2, 003 (2017)].

The new version of QuSpin features several important improvements, which I listed, in details, in the Strengths part of this report. In particular, the attention of authors to provide this many physically useful tools is admirable. However, it is important to note that most of these new features cannot be counted as major upgrades when we consider the physics of the targeted problems and the ED method itself, and certainly not worthy of a separate publication. Such updates are, of course, expected improvements for an evolving numerical package, the developers of which can be held accountable to make it more comprehensive and cost-efficient with the passage of time (see also below for a suggestion for listing the "updates" differently). On the other hands, the included step-by-step tutorials of physical models, although having significant pedagogical importance, and some are rather complicated to program and simulate from scratch but contain no new physics. Nevertheless, among the new features, I believe there are two truly major upgrades giving the paper the required originality: these are, of course, the addition of bases for higher than $S=\frac{1}{2}$ spins, fermions, and bosons, and providing the physical-basis support for a wide range of Hamiltonian symmetries. The aforesaid features are rarely available in other numerical toolkits (considering highly-efficient packages of all available modern numerical methods not only ED) and certainly require high efforts in programming and understanding the underlying physics.

As discussed, the new features of packages are introduced, in good details, through seven well-structured examples studying physical models, some of which are also being actively investigated on the frontier of modern physics. In the first example, the authors find the spectrum of the integrable transverse-field spin-$1/2$ ferromagnetic Ising chain using both the spins' computational basis and the
Jordan-Wigner transformation to switch to an interaction-free spinless fermionic Hamiltonian. I believe this example mainly teaches users how to construct spin and fermionic Hamiltonians, implement the boundary conditions, and avoid some common mistakes in doing so, and exploit spin inversion and reflection symmetries. In the next example, the characterizing gap in the spectrum and a non-equal-time correlator of the Su-Schrieffer-Heeger model are calculated. This example mainly teaches users how to construct free-particle translation-invariant Hamiltonians and time-evolve multiple quantum states simultaneously. In the third example, the authors study a dynamical phase transition (from a many-body-localised to an ergodic state) in the Fermi-Hubbard model with a random disorder. This example mainly teaches users how to construct Fock states and measure observables as a function of time efficiently. In the next example, the thermalization of an initial state of the non-integrable Bose-Hubbard model (placed on a short cylinder) is demonstrated through the study of entanglement entropy and local density of bosons. This example mainly teaches users how to construct a bosonic Hamiltonian on a ladder and to evolve a state over different symmetry sectors. In the fifth example, the authors (imaginary) time-evolve some states using Gross-Pitaevskii wave equations. This example mainly teaches users how to define time-dependent nonlinear equations of motion and try to solve such ODEs. In the next example, the dynamics of one-dimensional and square-lattice transverse-field Ising models are compared (mainly by studying some normalised relative energies) to show the non-integrability of the latter. This example mainly teaches users how to construct and define symmetries for the 2D systems. In their final tutorial, the authors study the dynamics of the entanglement entropy for a single species of the spinless Bose-Fermi mixture Hamiltonian. This tutorial mainly teaches users how to use tensor-product basis class and periodically time-evolve a state. In the end, it is worth mentioning that I was able to run all the provided example codes successfully.

Therefore, upon applying corrections in accordance with the required changes below, I strongly recommend the publication of this manuscript.

Requested changes

The following changes are required:

1- In Fig. 2(a), it is necessary to state (in the figure or the text) that the energies are in the unit of $J$. 2- In Page 9, the first paragraph, the original SSH papers [Phys. Rev. Lett. 42 1698-701 1979, Phys. Rev. B 22 2099-111 1980, Phys. Rev. B 28 1138(E) 1983], highlighting the history of the model and the relevance to polyacetylene, must be cited. It is suggested to also reference a modern take on the model to highlight the concept of emergent topological edge states. 3- It is not clear which numerical routines exp_op class uses to estimate the matrix exponentials. If this class also employs Python's expm function, as used in other places in the paper, the authors need to state this more clearly. If exp_op class exploit other numerical methods, the authors need to mention accordingly and add appropriate references. 4- In Page 2, the second paragraph, the authors correctly discuss that ED methods are still relevant to study certain dynamical problems. Another common and important use for ED calculations, in competition with highly-precise tensor network methods, is to derive the full or low-lying spectrum of frustrated Hamiltonians even on small system sizes. One needs to keep in mind that while employing tensor network approaches, although capable of studying very large system sizes, one requires to run a separate simulation to find the ground state and each individual excited state separately. I believe this other practical use of ED should be added to this paragraph. 5- In Pages 20 and 21, the discussed coefficients relate to Code Snippet 1, and not to Fig. 1 as stated in the text. 6- In Page 24, the sixth paragraph, the addition of at least one (original or review) reference covering Gross-Pitaevskii wave equations is necessary. Also, I like to suggest to put another reference covering the use of Gross-Pitaevskii equations, specifically, in nonlinear optics.

The following minor changes are suggested:

1- In Abstract, the authors appropriately specify the dimensions and/or lattice geometries of models in examples (iv) and (vi). The same can be also done for the models in other examples (e.g. using "Ising chain" instead of "Ising model"); especially, since mentioning the dimensionalities/geometries will reflect the degree of their complexity. 2- In Abstract, the SSH abbreviation is used for the first time here without stating the full version. Nevertheless, I suggest avoiding the use of the short form of the model name in the abstract. 4- In Page 3, the first paragraph, it is more appropriate to separate New Features into two groups as "major changes" and "other added features" (or any similar titles), as discussed in my Report, to highlight the importance of significant new changes marked as (i) and (ii). 5- The following is only a suggested change for the QuSpin core codes and not essentially relates the manuscript: as mentioned, the installation of the package is well-explained and rather straightforward by following the available guides and through the Python package manager Anaconda. However, the authors fail to address a seemingly common problem that I have faced during installation of QuSpin on several machines, i.e. the 'Permission denied' error while performing 'conda install -c weinbe58 quspin' (which perform several transactions at once). This would happen for any user with not enough permission in /address_to_anaconda directory and often cannot be fixed by changing umask settings (in addition, it is not recommended and useful to run anaconda with sudo). The solution is, of course, to simply run 'sudo chown user:user anaconda-dir/ -R' on all or relevant Anaconda directories. 6- There are no mentions of the available and important feature of Lanczos subspace diagonalization until the example in Sec. 2.5. It is suggested for this feature to be briefly discussed in the introduction. 7- The abbreviation GS, standing for the ground state, is first used in Page 27 without stating the full version. Nevertheless, I suggest avoiding the use of GS at any place. 8- Since the provided guideline for how to run Python codes is so brief (basically a one-liner as 'python test.py'), Appendix B seems very unnecessary. This one line can be then added to Appendix A. 9- Often when the authors discuss the spin-1/2 transverse-field Ising model on a chain (one-dimensional) or in two dimensions, they do not clearly state which models they really mean, e.g. see the last paragraph of Page 30. I encourage the authors to pay more attention to specifying the dimensionality of the Ising model. 10- In Sec. 2.6, it is never explicitly mentioned that the Ising model of the interest is on a square lattice. It would be useful to add this information to the beginning of the section. 11- The abbreviation DM, standing for the density matrix, is first used in Page 40 without stating the full version. Nevertheless, I suggest avoiding the use of DM at any place.

---

## Round 2 · Referee Report · Seyed Nariman Saadatmand · 2019-7-17

Report
In the resubmission, the authors accurately implement a number of changes to address most of the issues raised in my previous report, and also raised by the other referee. In a few minor cases, they have decided to leave the text as it was in the first version, however, their provided reasoning is satisfactory for me.
Author: Phillip Weinberg on 2019-08-01 [id 576]
(in reply to Report 2 on 2019-07-31)I have fixed the typos and replaced the preprint on Arxiv.

---

## Round 2 · Referee Report · Anonymous · 2019-7-31

Report
The authors have revised their paper according to the suggested points of improvements, and I can thus recommend publication of this manuscript in SciPost in its present form (after correcting the typos listed below).
Typos:
page 8 section title 2.2: "Su-Sschrieffer-Heeger" -> "Su-Schrieffer-Heeger"
page 42 second line: "GIthub" -> "Github"

---

## Round 2 · Author Response

List of changes
Report 1:
Suggested changes:
1. The only point of improvement I see is to better comment on the parallel capabilities of the package which is not very clear from the text (there is only one remark on p22, and a few remarks on p 40 with reference to the Documentation C). It would be interesting to have a quick overview somewhere, to what extend the package can be used in parallel computations. (If certain parts do not support parallelism yet, that could be mentioned as possible feature in a future release).
We added an explicit page, and a corresponding example, to the online documentation which explains how to use the parallel capabilities. Additionally, starting from version 0.3.2, QuSpin has full OpenMP support. Because the paper is anyway long, we only mention the parallel capabilities in the Outlook section, and refer the interested readers to the online documentation for an example how to use them.
2. a few typos and minor points:
- p.8 title of Sec 2.2: write out SSH -> Su-Schrieffer-Heeger
- Figure 4 caption rhs: (b) local -> (b) Local
- p.24 after the first code block: adder -> ladder
- Fig.5: figure labels are borderline small
- p27, after code line 41: Hsmiltonian -> Hamiltonian
- p41, first paragraph: Linblad -> Lindblad
We corrected these typos.
Report 2:
Requested changes
The following changes are required:
1- In Fig. 2(a), it is necessary to state (in the figure or the text) that the energies are in the unit of J.
We added the clarification to the caption.
2- In Page 9, the first paragraph, the original SSH papers [Phys. Rev. Lett. 42 1698-701 1979, Phys. Rev. B 22 2099-111 1980, Phys. Rev. B 28 1138(E) 1983], highlighting the history of the model and the relevance to polyacetylene, must be cited. It is suggested to also reference a modern take on the model to highlight the concept of emergent topological edge states.
We added the references.
3- It is not clear which numerical routines exp_op class uses to estimate the matrix exponentials. If this class also employs Python's expm function, as used in other places in the paper, the authors need to state this more clearly. If exp_op class exploit other numerical methods, the authors need to mention accordingly and add appropriate references.
We added a footnote in Sec 2.2. to clarify this point.
4- In Page 2, the second paragraph, the authors correctly discuss that ED methods are still relevant to study certain dynamical problems. Another common and important use for ED calculations, in competition with highly-precise tensor network methods, is to derive the full or low-lying spectrum of frustrated Hamiltonians even on small system sizes. One needs to keep in mind that while employing tensor network approaches, although capable of studying very large system sizes, one requires to run a separate simulation to find the ground state and each individual excited state separately. I believe this other practical use of ED should be added to this paragraph.
We added this example to the discussion.
5- In Pages 20 and 21, the discussed coefficients relate to Code Snippet 1, and not to Fig. 1 as stated in the text.
We corrected this typo.
6- In Page 24, the sixth paragraph, the addition of at least one (original or review) reference covering Gross-Pitaevskii wave equations is necessary. Also, I like to suggest to put another reference covering the use of Gross-Pitaevskii equations, specifically, in nonlinear optics.
We added the requested references.
The following minor changes are suggested:
1- In Abstract, the authors appropriately specify the dimensions and/or lattice geometries of models in examples (iv) and (vi). The same can be also done for the models in other examples (e.g. using "Ising chain" instead of "Ising model"); especially, since mentioning the dimensionalities/geometries will reflect the degree of their complexity.
We included these clarifications.
2- In Abstract, the SSH abbreviation is used for the first time here without stating the full version. Nevertheless, I suggest avoiding the use of the short form of the model name in the abstract.
We agree to this comment and corrected the sentence.
4- In Page 3, the first paragraph, it is more appropriate to separate New Features into two groups as "major changes" and "other added features" (or any similar titles), as discussed in my Report, to highlight the importance of significant new changes marked as (i) and (ii).
We decided to leave the text as it is, mainly because we keep adding new features to the package. To keep track of them, we now have a separate page on the documentation website, where we divide the newly introduced changes according to Improved Functionality and New Classes, Methods, and Functions.
5- The following is only a suggested change for the QuSpin core codes and not essentially relates the manuscript: as mentioned, the installation of the package is well-explained and rather straightforward by following the available guides and through the Python package manager Anaconda. However, the authors fail to address a seemingly common problem that I have faced during installation of QuSpin on several machines, i.e. the 'Permission denied' error while performing 'conda install -c weinbe58 quspin' (which perform several transactions at once). This would happen for any user with not enough permission in /address_to_anaconda directory and often cannot be fixed by changing umask settings (in addition, it is not recommended and useful to run anaconda with sudo). The solution is, of course, to simply run 'sudo chown user:user anaconda-dir/ -R' on all or relevant Anaconda directories.
In the online documentation, under Installation, we have put a paragraph on "Installing without sudo Privileges". The simplest way of doing it using anaconda is to create a new environment, where no 'sudo'-commands are required. If this fails, the user can install miniconda (a light but sufficient version of anaconda) in their home directory where they typically do have rights to install software programs, and then use miniconda.
6- There are no mentions of the available and important feature of Lanczos subspace diagonalization until the example in Sec. 2.5. It is suggested for this feature to be briefly discussed in the introduction.
While Lanczos is a substantial part of any ED method, we do not feel that we contribute in any way to these algorithms/methods. As we acknowledge throughout the text, we make use of Scipy routines for these Krylov methods and cite the relevant libraries. Scipy provides a decent discussion of these methods in their documentation. Since the paper is anyway long enough, we prefer to abstain from this discussion.
7- The abbreviation GS, standing for the ground state, is first used in Page 27 without stating the full version. Nevertheless, I suggest avoiding the use of GS at any place.
We incorporated this suggestion.
8- Since the provided guideline for how to run Python codes is so brief (basically a one-liner as 'python test.py'), Appendix B seems very unnecessary. This one line can be then added to Appendix A.
We agree to this point. Yet, we believe that there is no harm in leaving these instructions for the benefit of new-comers to python.
9- Often when the authors discuss the spin-1/2 transverse-field Ising model on a chain (one-dimensional) or in two dimensions, they do not clearly state which models they really mean, e.g. see the last paragraph of Page 30. I encourage the authors to pay more attention to specifying the dimensionality of the Ising model.
We clarified the last paragraph of Page 30 and fixed the wording to make it less confusing.
10- In Sec. 2.6, it is never explicitly mentioned that the Ising model of the interest is on a square lattice. It would be useful to add this information to the beginning of the section.
We added this information to the text.
11- The abbreviation DM, standing for the density matrix, is first used in Page 40 without stating the full version. Nevertheless, I suggest avoiding the use of DM at any place.
We removed this abbreviation.

---

## Round 2 · List of Changes

Report 1:
Suggested changes:
1. The only point of improvement I see is to better comment on the parallel capabilities of the package which is not very clear from the text (there is only one remark on p22, and a few remarks on p 40 with reference to the Documentation C). It would be interesting to have a quick overview somewhere, to what extend the package can be used in parallel computations. (If certain parts do not support parallelism yet, that could be mentioned as possible feature in a future release).
We added an explicit page, and a corresponding example, to the online documentation which explains how to use the parallel capabilities. Additionally, starting from version 0.3.2, QuSpin has full OpenMP support. Because the paper is anyway long, we only mention the parallel capabilities in the Outlook section, and refer the interested readers to the online documentation for an example how to use them.
2. a few typos and minor points:
- p.8 title of Sec 2.2: write out SSH -> Su-Schrieffer-Heeger
- Figure 4 caption rhs: (b) local -> (b) Local
- p.24 after the first code block: adder -> ladder
- Fig.5: figure labels are borderline small
- p27, after code line 41: Hsmiltonian -> Hamiltonian
- p41, first paragraph: Linblad -> Lindblad
We corrected these typos.
Report 2:
Requested changes
The following changes are required:
1- In Fig. 2(a), it is necessary to state (in the figure or the text) that the energies are in the unit of J.
We added the clarification to the caption.
2- In Page 9, the first paragraph, the original SSH papers [Phys. Rev. Lett. 42 1698-701 1979, Phys. Rev. B 22 2099-111 1980, Phys. Rev. B 28 1138(E) 1983], highlighting the history of the model and the relevance to polyacetylene, must be cited. It is suggested to also reference a modern take on the model to highlight the concept of emergent topological edge states.
We added the references.
3- It is not clear which numerical routines exp_op class uses to estimate the matrix exponentials. If this class also employs Python's expm function, as used in other places in the paper, the authors need to state this more clearly. If exp_op class exploit other numerical methods, the authors need to mention accordingly and add appropriate references.
We added a footnote in Sec 2.2. to clarify this point.
4- In Page 2, the second paragraph, the authors correctly discuss that ED methods are still relevant to study certain dynamical problems. Another common and important use for ED calculations, in competition with highly-precise tensor network methods, is to derive the full or low-lying spectrum of frustrated Hamiltonians even on small system sizes. One needs to keep in mind that while employing tensor network approaches, although capable of studying very large system sizes, one requires to run a separate simulation to find the ground state and each individual excited state separately. I believe this other practical use of ED should be added to this paragraph.
We added this example to the discussion.
5- In Pages 20 and 21, the discussed coefficients relate to Code Snippet 1, and not to Fig. 1 as stated in the text.
We corrected this typo.
6- In Page 24, the sixth paragraph, the addition of at least one (original or review) reference covering Gross-Pitaevskii wave equations is necessary. Also, I like to suggest to put another reference covering the use of Gross-Pitaevskii equations, specifically, in nonlinear optics.
We added the requested references.
The following minor changes are suggested:
1- In Abstract, the authors appropriately specify the dimensions and/or lattice geometries of models in examples (iv) and (vi). The same can be also done for the models in other examples (e.g. using "Ising chain" instead of "Ising model"); especially, since mentioning the dimensionalities/geometries will reflect the degree of their complexity.
We included these clarifications.
2- In Abstract, the SSH abbreviation is used for the first time here without stating the full version. Nevertheless, I suggest avoiding the use of the short form of the model name in the abstract.
We agree to this comment and corrected the sentence.
4- In Page 3, the first paragraph, it is more appropriate to separate New Features into two groups as "major changes" and "other added features" (or any similar titles), as discussed in my Report, to highlight the importance of significant new changes marked as (i) and (ii).
We decided to leave the text as it is, mainly because we keep adding new features to the package. To keep track of them, we now have a separate page on the documentation website, where we divide the newly introduced changes according to Improved Functionality and New Classes, Methods, and Functions.
5- The following is only a suggested change for the QuSpin core codes and not essentially relates the manuscript: as mentioned, the installation of the package is well-explained and rather straightforward by following the available guides and through the Python package manager Anaconda. However, the authors fail to address a seemingly common problem that I have faced during installation of QuSpin on several machines, i.e. the 'Permission denied' error while performing 'conda install -c weinbe58 quspin' (which perform several transactions at once). This would happen for any user with not enough permission in /address_to_anaconda directory and often cannot be fixed by changing umask settings (in addition, it is not recommended and useful to run anaconda with sudo). The solution is, of course, to simply run 'sudo chown user:user anaconda-dir/ -R' on all or relevant Anaconda directories.
In the online documentation, under Installation, we have put a paragraph on "Installing without sudo Privileges". The simplest way of doing it using anaconda is to create a new environment, where no 'sudo'-commands are required. If this fails, the user can install miniconda (a light but sufficient version of anaconda) in their home directory where they typically do have rights to install software programs, and then use miniconda.
6- There are no mentions of the available and important feature of Lanczos subspace diagonalization until the example in Sec. 2.5. It is suggested for this feature to be briefly discussed in the introduction.
While Lanczos is a substantial part of any ED method, we do not feel that we contribute in any way to these algorithms/methods. As we acknowledge throughout the text, we make use of Scipy routines for these Krylov methods and cite the relevant libraries. Scipy provides a decent discussion of these methods in their documentation. Since the paper is anyway long enough, we prefer to abstain from this discussion.
7- The abbreviation GS, standing for the ground state, is first used in Page 27 without stating the full version. Nevertheless, I suggest avoiding the use of GS at any place.
We incorporated this suggestion.
8- Since the provided guideline for how to run Python codes is so brief (basically a one-liner as 'python test.py'), Appendix B seems very unnecessary. This one line can be then added to Appendix A.
We agree to this point. Yet, we believe that there is no harm in leaving these instructions for the benefit of new-comers to python.
9- Often when the authors discuss the spin-1/2 transverse-field Ising model on a chain (one-dimensional) or in two dimensions, they do not clearly state which models they really mean, e.g. see the last paragraph of Page 30. I encourage the authors to pay more attention to specifying the dimensionality of the Ising model.
We clarified the last paragraph of Page 30 and fixed the wording to make it less confusing.
10- In Sec. 2.6, it is never explicitly mentioned that the Ising model of the interest is on a square lattice. It would be useful to add this information to the beginning of the section.
We added this information to the text.
11- The abbreviation DM, standing for the density matrix, is first used in Page 40 without stating the full version. Nevertheless, I suggest avoiding the use of DM at any place.
We removed this abbreviation.

---

## Editorial Decision

published